# Distinct interacting cortical networks for stimulus-response and repetition-suppression

David Eckert[1,2], Christoph Reichert [2], Christian G. Bien[3], Hans-Jochen Heinze[1,2,4,5,6], Robert T. Knight[7,8], Leon Y. Deouell[9] & Stefan Dürschmid[1,2 ✉]

Non-invasive studies consider the initial neural stimulus response (SR) and repetition suppression (RS) – the decreased response to repeated sensory stimuli – as engaging the same neurons. That is, RS is a suppression of the SR. We challenge this conjecture using electrocorticographic (ECoG) recordings with high spatial resolution in ten patients listening to task-irrelevant trains of auditory stimuli. SR and RS were indexed by high-frequency activity (HFA) across temporal, parietal, and frontal cortices. $HFA_{SR}$ and $HFA_{RS}$ were temporally and spatially distinct, with $HFA_{RS}$ emerging later than $HFA_{SR}$ and showing only a limited spatial intersection with $HFA_{SR}$: most $HFA_{SR}$ sites did not demonstrate $HFA_{RS}$, and $HFA_{RS}$ was found where no $HFA_{SR}$ could be recorded. β activity was enhanced in $HFA_{RS}$ compared to $HFA_{SR}$ cortical sites. θ activity was enhanced in $HFA_{SR}$ compared to $HFA_{RS}$ sites. Furthermore, $HFA_{SR}$ sites propagated information to $HFA_{RS}$ sites via transient θ:β phase-phase coupling. In contrast to predictive coding (PC) accounts our results indicate that $HFA_{SR}$ and $HFA_{RS}$ are functionally linked but have minimal spatial overlap. $HFA_{SR}$ might enable stable and rapid perception of environmental stimuli across extended temporal intervals. In contrast $HFA_{RS}$ might support efficient generation of an internal model based on stimulus history.

[1] Department of Neurology, Otto-von-Guericke University of Magdeburg, Leipziger Str. 44, 39120 Magdeburg, Germany. [2] Department of Behavioral Neurology, Leibniz Institute for Neurobiology, Brenneckestr. 6, 39120 Magdeburg, Germany. [3] Department. of Epileptology, Krankenhaus Mara, Bielefeld University, Maraweg 21, 33617 Bielefeld, Germany. [4] Forschungscampus STIMULATE, Otto-von-Guericke University of Magdeburg, Universitätsplatz 2, 39106 Magdeburg, Germany. [5] CBBS - center of behavioral brain sciences, Otto-von-Guericke University of Magdeburg, Universitätsplatz 2, 39106 Magdeburg, Germany. [6] German Center for Neurodegenerative Diseases (DZNE), Leipziger Str. 44, 39120 Magdeburg, Germany. [7] Department of Psychology, University of California Berkeley, 130 Barker Hall, Berkeley 94720 CA, USA. [8] Helen Wills Neuroscience Institute, University of California Berkeley, Berkeley 94720 CA, USA. [9] Department of Psychology and Edmond and Lily Safra Center for brain sciences, The Hebrew University of Jerusalem, Jerusalem, Israel. ✉email: stefan.duerschmid@lin-magdeburg.de

A ubiquitous finding in neuroscience is that neural responses to repeated stimuli are reduced compared to initial stimulus presentation, the phenomenon of repetition suppression (RS). RS has been shown in both single-unit studies in the monkey cortex[1–3] as well as noninvasive studies in humans using different techniques (for a review, see ref. [4]). Several explanations for RS have been put forward, including adaptation or habituation, sharpening of representations, and reduction of prediction errors[4–12]. This reduction of responses to frequently occurring stimuli is associated with an enhanced response to unexpected events, establishing a mechanism for change detection[13,14], with the probability of stimulus events accounting for a large proportion of neural variability[9] Most hypotheses on the mechanisms responsible for RS assume that what is suppressed is the stimulus-induced response. That is, the same neurons or networks that are initially responsive to the stimulus are the ones which are less active when the same stimulus repeats. Noninvasive studies in humans report that RS and stimulus response (SR) overlap, but these methods cannot distinguish nearby cortical activity.

A critical question remains whether RS is restricted to reducing the SR, in which case SR and RS should co-occur in the same electrodes (henceforward SR+RS+ sites), as suggested by scalp EEG and MEG recordings. Alternatively, SR and RS could be dissociated, but the circuits exhibiting SR and RS are intermingled and not resolvable by low-resolution scalp recording. This potential dissociation could be measured using a direct cortical recording of broadband high-frequency activity (HFA, 80–150 Hz), which is the key response frequency in previous ECoG (electrocorticography) studies[15–19] studying SR and RS.

Here, we utilized the high temporal and spatial resolution of direct cortical recordings from subdural ECoG electrodes to compare SR+ and RS+ signals in ten patients presented with trains of task-irrelevant auditory stimuli while attending a visual slide show to probe the automatic nature of RS. We show that while SR and RS both engage frontal, parietal, and temporal regions, they can be dissociated temporally and spatially in HFA. Critically, HFA SR+ and RS+ sites are distinctly modulated by θ and β low-frequency activity, respectively, with mutual information flow from SR+ to RS+ sites.

## Results

For easy reference, the results sections correspond to the similarly enumerated sections of the methods section.

**I – Stimulus response.** We studied 412 channels (95 channels over frontal, 202 channels over temporal, and 115 channels over parietal regions across all subjects (see Fig. 1b)). Ninety-one channels showed a significant HFA modulation to auditory tones (of which 84 showed a stimulus response without RS, SR+RS−; Fig. 1b, c). Stimulus response occurred between 17–393 ms ($SR_{max}$ at 113 ms, $p < 0.001$) over multiple cortical regions (11 channels over frontal, 53 over temporal, and 20 over parietal regions, see Fig. 1b and Table 1), manifested as an increase in HFA power. None of the RS+ channels in our study showed a significant reduction of the HFA relative to the pre-stimulus period in response to the first standard.

**II – Repetition suppression.** RS (defined as both a significant $F_{N\ standard}$ and significant negative trend ($r$ value) from S1 to S3) was found in 31 channels (of which 24 channels did not show SR, SR−RS+ Fig. 1d and Table 1) between 48–436 ms following stimulus onset ($F_{max}$ at 265 ms, $p < 0.001$, Fig. 1e). Channels designated as SR−RS+ showed no trend towards a stimulus response (average $p = 0.41$, SD = 0.1) even with a

liberal, uncorrected significance criteria. We found three channels showing RE (both a significant $F_{N\ standard}$ and significant positive $r$ value), in the parietal ($N = 2$) and the temporal ($N = 1$) cortex. None of these channels showed significant SR (i.e., they were SR−RE+ channels).

**III – Comparison of stimulus response and repetition suppression.** About 115 channels exhibited SR and/or RS. Of these, only seven channels showed both (designated SR+RS+). The remaining showed only SR (SR+RS−, 84 channels, Fig. 2a) or showed RS without SR (SR−RS+; 24 channels, Fig. 2b). We first confirmed that the lack of SR in SR−RS+ sites is not due to reduced sensitivity caused by high baseline variance ($ς^2$) in these channels compared to other sites (baseline −200– 0 ms, $F_{(3,408)} = 0.37$; $p = 0.78$; SR+RS−: mean $ς^2 = 0.0012$, std = 0.004; SR−RS+: $ς^2 = 0.0008$, std = 0.0008; SR+RS+: $ς^2 = 0.0017$, std = 0.002; $ς^2 = 0.0009$, std = 0.0037). We also estimated the BF to determine the amount of evidence for a change over baseline separately for each channel at each time point. Bayes factor analysis provided strong support to lack of stimulus response in SR−RS+ channels ($BF_{mean} = 0.12$, $BF_{min} = 0.098$, $BF_{max} = 0.19$; Fig. 2c), and SR−RS− channels ($BF_{mean} = 0.17$, $BF_{min} = 0.15$, $BF_{max} = 0.21$), while strong evidence for stimulus response was observed in SR+RS+ channels ($BF_{mean} = 570.03$, $BF_{min} = 289.7$, $BF_{max} = 7590$), and in SR+RS− channels ($BF_{mean} = 54.54$, $BF_{min} = 34.5$, $BF_{max} = 506.19$). We additionally estimated the BF to determine the amount of evidence for repetition suppression separately for each channel. We estimated the RS effect in each S1, S2, and S3 sequence as the summed difference between HFA to S1, S2, and S3 (response following S3 vs response following $S_2$ and response following $S_2$ vs response following $S_1$, respectively). This gives a difference wave for each sequence presented. We found support for no repetition suppression, neither in SR+RS− ($BF_{mean} = 0.23$, $BF_{min} = 0.21$, $BF_{max} = 0.26$) nor SR−RS− channels ($BF_{mean} = 0.23$, $BF_{min} = 0.22$, $BF_{max} = 0.25$). However, we observed positive evidence in SR+RS+ channels ($BF_{mean} = 315.72$, $BF_{min} = 172.7$, $BF_{max} = 1928.4$) and in SR−RS+ channels ($BF_{mean} = 16.52$, $BF_{min} = 4.49$, $BF_{max} = 376$; Fig. 2d). Temporal parameters also distinguished SR and RS channels. We found that RS peaked significantly later than SR in SR+RS+ ($SR_{peak} = 158$ ms, $RS_{peak} = 230$ ms, $t_6 = 2.55$; $p = 0.04$, Fig. 2e) and all SR+RS− and SR−RS+ channels combined ($SR_{peak} = 167$ ms, $RS_{peak} = 253$ ms, $t_{106} = 3.62$; $p < 0.001$, Fig. 2e).

**IV – Comparison of dominant band power.** We then asked whether SR and RS sites dissociate in spectral characteristics. Power spectral-density (PSD) showed an interaction between factors Channel Type (SR+RS−, SR−RS+, SR−RS−, SR+RS+) and Frequency Bands (θ, α, β) ($F_{6,1224} = 3.2$; $p = 0.013$; Fig. 3a). θ and β activity differed significantly between channel types but not α activity, see Table 2. Post hoc tests revealed stronger θ power ($P_θ$) in SR+ than SR− channels ($P_{θ\_SR+RS+} < P_{θ\_SR+RS−}$; $t_{106} = 2.51$; $q = 0.04$; $P_{θ\_SR+RS−} > P_{θ\_SR−RS−}$; $t_{379} = 2.11$; $q = 0.052$ and a trend towards $P_{θ\_SR+RS+} > P_{θ\_SR+RS+}$; $t_{29} = 1.50$; $q = 0.082$; $t_{crit} = 1.99$ denotes the $t$ value which the observed $t$ values had to exceed to be considered significant). In contrast, β activity showed stronger power for RS+ than RS− channels ($P_{θ\_SR−RS+} > P_{β\_SR+RS−}$, $t_{106} = 2.37$, $q = 0.0297$; $P_{θ\_SR−RS+} > P_{θ\_SR−RS−}$, $t_{319} = 2.41$, $q = 0.0297$; and a trend towards $P_{β\_SR+RS−} < P_{β\_SR+RS+}$, $t_{89} = 1.5$, $q = 0.082$ and $P_{β\_SR−RS−} < P_{β\_SR+RS+}$, $t_{302} = 1.4$, $q = 0.084$; Fig. 3b). These results show higher β power in SR−RS+ channels than SR+RS− channels—despite a small reduction in β power from $S_1$ through $S_3$, and stronger θ power in SR+RS−channels show than SR−RS+ channels even though the latter exhibit numerically repetition enhancement. (see Fig. 3b right panel).

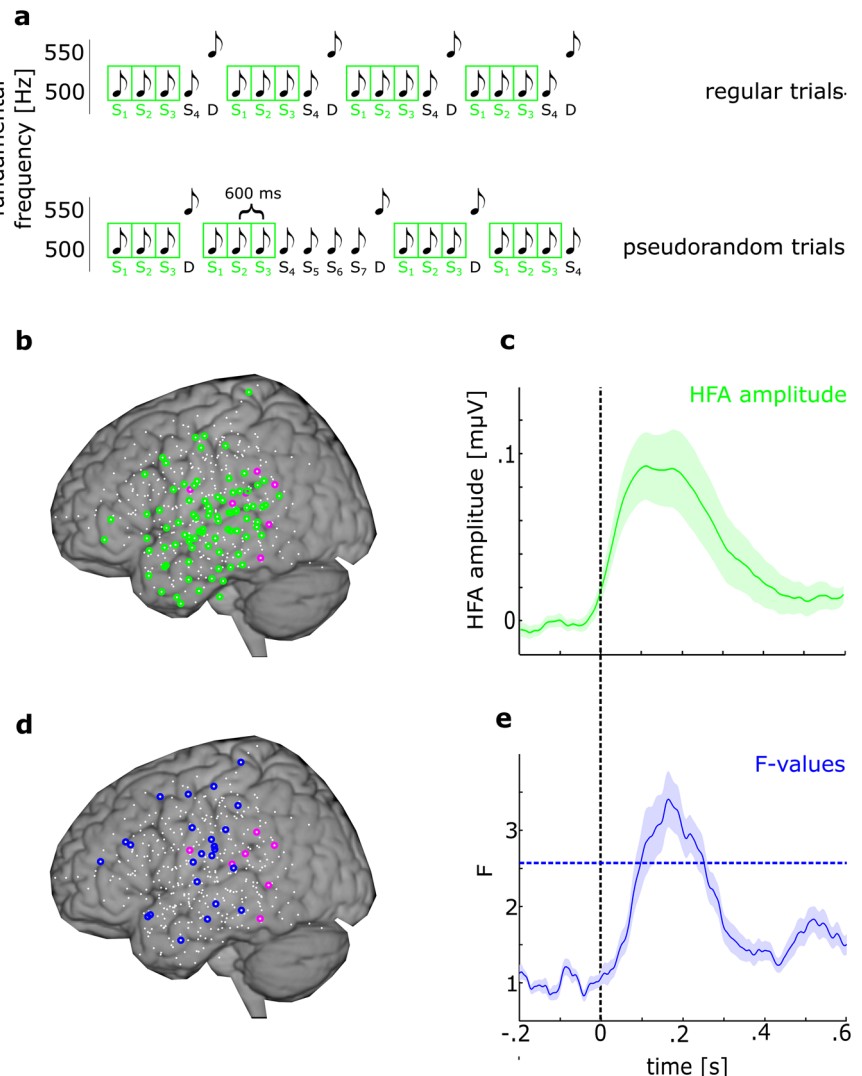

**Fig. 1 Stimulus response and repetition suppression show different spatial profiles. a** shows the auditory oddball paradigm. While the occurrence of deviants was unpredictable, $S_1$, $S_2$, and $S_3$ were always predictable. **b** shows the spatial distribution of HFA SR+RS− (green) and SR+RS+ (magenta) channels as marked dots against the background of a standard schematic brain using MNI coordinates that was also used for surgical planning. The remaining electrodes are marked by small white dots. **c** shows the HFA amplitude modulation of SR+ channels over time averaged across electrodes. The shaded area denotes the standard error across channels. **d** shows the spatial distribution of HFA SR−RS+ channels (blue) analog to **b**. Channels with magenta circles represent SR+RS+, as in **b**. **e** Modulation of repetition-suppression (*F* values) in SR−RS+ channels. The dashed blue line represents the significance threshold of *F* values. The shaded area denotes the standard error across the channels.

| Table 1 Summary of channels. | | | |
|---|---|---|---|
| **Patient** | **HFA SR-RS+ channels (temporal/frontal/parietal)** | **HFA SR+RS- channels (temporal/frontal/parietal)** | **Total no. of electrodes** |
| I | 3 (1/2/0) | 22 (13/5/4) | 60 |
| II | 3 (0/0/3) | 17 (8/1/8) | 59 |
| III | 5 (2/0/3) | 11 (8/1/2) | 52 |
| IV | 4 (2/0/2) | 2 (0/1/1) | 56 |
| V | 2 (1/1/0) | 13 (7/3/3) | 60 |
| VI | 1 (0/0/1) | 3 (2/0/1) | 15 |
| VII | 3 (1/0/2) | 5 (4/0/1) | 26 |
| VIII | 1 (1/0/0) | 0 (0/0/0) | 16 |
| IX | 2 (0/2/0) | 2 (2/0/0) | 53 |
| X | 0 (0/0/0) | 9 (9/0/0) | 15 |
| total No | 24 (8/5/11) | 84 (53/11/20) | 412 |

**V – Cross-frequency coupling**. Within channel low frequency:HFA phase:amplitude coupling (PAC, Fig. 3c, d) showed no main effects of channel type or frequency band ($F_{\text{channel type}} = 0.28$; $p = 0.59$, $F_{\text{frequency band}} = 1.5$; $p = 0.22$), but a significant interaction ($F = 4.79$; $p = 0.03$), reflecting stronger SR–RS+ HFA coupling to β than θ phase and vice versa in SR+RS− channels (Fig. 3e).

**VI – SR+:RS+ integration**. The distinct SR and RS cortical sites raise the question of whether SR+RS− and SR–RS+ sites interact. SR+RS− and SR–RS+ electrodes showed increased post-stimulus (116–270 ms) phase:phase coupling of $\theta_{\text{SR+}}$ and $\beta_{\text{RS+}}$ ($\kappa_{\text{crit}} = 0.0043$; $\kappa_{\text{max}} = 0.019$; $p < 0.00001$, Fig. 3f) indicating significant interaction (Fig. 3g, h). Phase concentration coefficient κ differed significantly across frequency band pairs between 160–261 ms ($F_{\text{crit}} = 3.95$ denotes the critical *F* value which the observed *F* values had to exceed to be considered significant; max $F_{2,288} = 8.27$; $p < 0.00001$), due to stronger θ:β coupling ($\kappa_{\theta:\beta} = 0.0091$) than θ:α

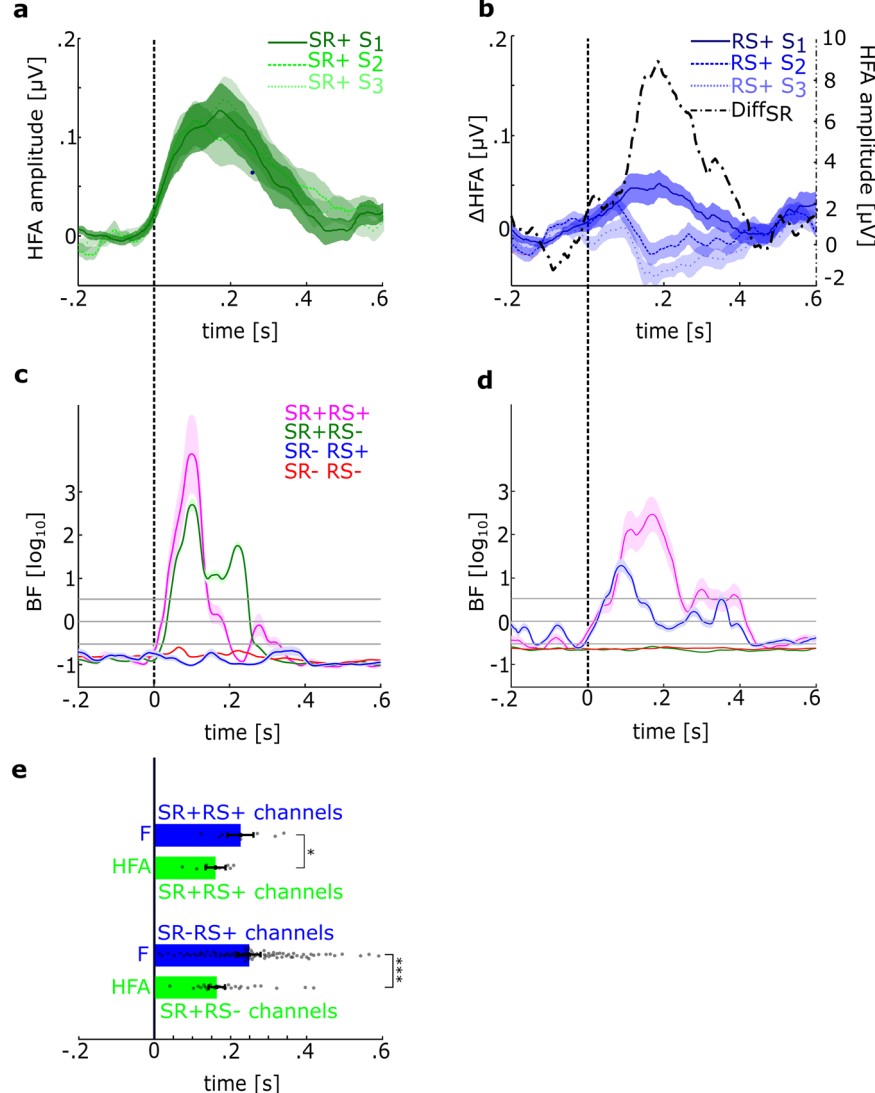

**Fig. 2 Stimulus response and repetition suppression show different temporal profiles. a**, **b** show the HFA amplitude modulation of SR+RS– (**a**) and SR–RS+ (**b**) channels to $S_1$, $S_2$, and $S_3$, the dashed black curve in **b** represents the summed sign of differences between $S_1$ & $S_2$ and $S_2$ & $S_3$ (Diff$_{SR}$). Positive values of this summed differences indicate a $S_1 > S_2 > S_3$ relationship, shaded areas represent the standard error of the mean. **c**, **d** show time-resolved BF for modulation of HFA over baseline (evidence for stimulus response) averaged separately across the four different channel sets (SR–RS+, SR+RS–, SR–RS–, and SR+RS+) (**c**) and for the effect of repetition suppression averaged across channels, shaded areas represent the standard error of the mean (**d**). Only SR+RS– and SR+RS+ channels show evidence for a stimulus response, only SR–RS+ and SR+RS+ channels show evidence for RS. The gray horizontal lines correspond to Bayes values 3, 1, and 0.3 with 3 and 0.3 denoting the critical levels for evidence for the alternate and null hypothesis, respectively. **e** shows peak latencies for RS (*F* values, blue) and SR (HFA, green), compared within SR+RS+ channels (upper bars) and between SR+RS– and SR–RS+ channels (lower bars), error bars represent the standard error of the mean, gray dots represent single data.

coupling ($\kappa_{\theta:\alpha} = 0.0004$; $t = 3.25$; $p = 0.0016$), and $\alpha:\beta$ coupling ($\kappa_{\alpha:\beta} = 0.0009$; $t = 5.03$; $p < 0.0001$ Fig. 3i, j).

**VII – Information propagation**. Between-sites mutual information (MI) analysis revealed early SR+ activity between 89–192 ms to be predictive of later RS+ activity between 190–226 ms (MI$_{crit}$ = 0.85 bits; MI$_{max}$ = 0.86 bits at 141 ms of SR+ and 210 ms of RS+ time series; $p < 0.00001$. Fig. 3k) suggesting information propagation from SR+ to RS+ sites.

## Discussion

Numerous studies report that the response to sensory stimuli decreases with repeating stimulation, a phenomenon known as repetition suppression (RS) or stimulus-specific adaptation[14]. Noninvasive studies report substantial spatial overlap of stimulus

response and repetition suppression, but such studies are limited in spatial resolution[20]. Thus, these methods are not well suited to examine whether RS reflects a stimulus response which gets reduced upon repeating stimulation, or might be a separate phenomenon of activity reduction relative to baseline in sites lacking stimulus response. This type of RS might reflect a short-term memory mechanism, independent of stimulus response. Here, we used intracranial EEG data in the context of repeating tones to measure the temporal, spatial, and spectral features of both phenomena. Unlike previous studies studying RS, we did not limit our analyses to channels showing stimulus response. If RS depended on sites being responsive to the stimuli, we should expect spatial overlap between the two, yet we found many sites showing exclusively SR and sites showing exclusively RS. In fact, a surprisingly small proportion of sites showed both SR and RS. Moreover, SR and RS exclusive sites also showed distinct spectral

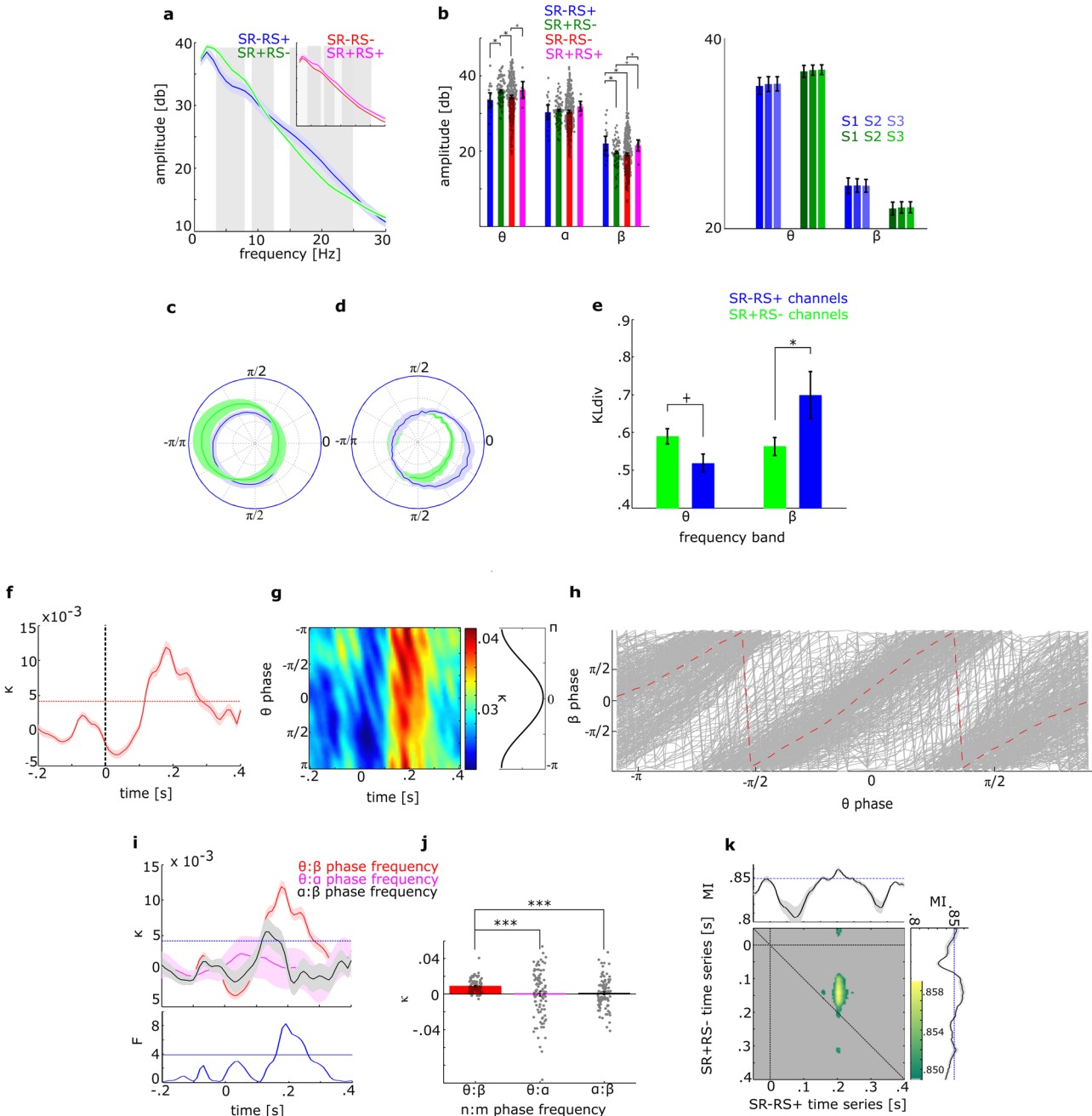

**Fig. 3 Stimulus response and repetition suppression show complex spectral interactions. a** Power spectral density (PSD) for SR–RS+ and SR+RS– channels (inset shows PSD for channels showing both/neither effect). **b** Spectral power averaged within three canonical frequency bands, showing higher β power in RS+ (SR+RS+, SR–RS+) than RS– (SR+RS–, SR–RS–) channels and higher θ power in SR+ (SR+RS–, SR+RS+) compared to SR– (SR–RS+, SR–RS–) channels. No differences between channel sets were found within the α band (unsaturated color bars), gray dots represent single data. The right panel shows spectral power for the three consecutive standard trials separately, for the SR–RS+ channels (blue) and SR+RS- channels (green) separately. **c, d** HFA modulation by θ (**c**) and β (**d**) for SR–RS+ (blue) and SR+RS– (green) channels. **e** Kullback–Leibler divergence of θ and β for SR–RS+ and SR+RS–. **f, g** modulation of concentration index κ, indicating transient θ:β interaction, averaged across phases (**f**, the dashed horizontal red line shows the significance threshold) and for each θ phase (**g**). **h** θ-phase:β-phase coupling at $κ_{max}$, gray lines represent pairwise combinations, red dashed line shows the average across all pairs. **i** the κ-comparison between θ:β, θ:α, and α:β (bottom: ANOVA F values). **j** Post hoc comparisons of θ:β, θ:α, and α:β, gray dots represent single data. (*** indicates $p < 0.001$) **k** Directed information propagation from SR+RS– to SR–RS+. In all panels, the shaded error margin around lines indicates the standard error across channels.

characteristics. SR sites showed higher θ power and θ:HFA phase–amplitude coupling (PAC) than sites with no SR, and RS sites showed stronger β power and stronger β:HFA PAC than sites showing no RS. Finally, we show that the two processes interact - the two types of sites show phase-phase coupling between their dominant theta and beta frequencies. Further, HFA

SR peaks earlier than HFA RS effects, and HFA SR predicts HFA RS.

Stimulus response was evident in all three lobes measured (frontal, parietal and temporal), manifested in high-frequency activity. All three regions also showed RS, with an amplitude decrease to repeated stimulation. Previous ECoG studies have

**Table 2 Comparison of PSDs between channel types SR+RS-, SR-RS+, SR-RS-, and SR+RS+ channels, respectively, each for the three frequency bands θ, β, and α based on the interaction between both factors.**

| | | Power spectral-density (PSD) | | | | F | p |
|---|---|---|---|---|---|---|---|
| | | SR+RS- | SR-RS+ | SR-RS- | SR+RS+ | | |
| frequency band | θ | 35.9 | 33.4 | 34.3 | 36.1 | $F_{3,408} = 3.37$ | 0.012 |
| | β | 19.5 | 21.9 | 19.1 | 21.5 | $F_{3,408} = 2.76$ | 0.04 |
| | α | 30.7 | 30.2 | 30.5 | 31.5 | $F_{3,408} = 0.3$ | 0.82 |

shown adaptation of high-frequency activity, mostly located in the temporal and parietal cortex[21], and here we show that lateral frontal cortex sites adapt to frequent auditory repetition as well, which has only been shown for low frequencies[22,23].

What could be the mechanisms driving SR with no apparent RS? Under the traditional explanation of RS as a process of adaptation which recovers with time[24], SR without RS might reflect sites with recovery times shorter than our inter-stimulus intervals, such that full recovery has been attained. Indeed, previous findings suggested variability in adaptation time constants[24]. The time constant of neuronal populations can be seen as reflecting the temporal resolution of the representation of the environment. Differences in resolution might allow distinguishing processing of coarse and fine-grained details[24]. Previous studies in vision show that the phase of θ activity across frontal and parietal regions is related to rhythmic sampling predicting visual detection performance[25,26]. Here, HFA SR+ modulated by θ activity (as evidenced by an increase in PAC), independent of RS, could be a mechanism to detect sensory evidence independent of context information or expectations.

An alternative explanation suggested by a reviewer is that SR–RS+ channels reflect the summed response of neurons excited by the stimulus and show repetition suppression (i.e. genuine SR+RS+) and of neurons that are a-priori inhibited by the stimulus. These inhibited neurons would "cancel out" the apparent SR in our mesoscopic recording of LFPs, making it look as if RS is present without SR. Without extensive single neuron measurements, it is difficult to rule out this possibility. However, we suggest that this possibility is less likely, considering the different spectral characteristics of SR+ sites and RS+ sites. If apparent SR–RS+ channels represent a composite signal of genuine SR+RS+ and inhibited neurons, then SR–RS+ channels should show comparable spectral signatures as other SR+ channels (namely SR+RS- and SR+RS+). However, both in the θ and the β range, SR–RS+ channels show a spectral signature which resembles other SR- channels (SR–RS−) and differs from SR+ channels. Specifically, in the θ range, SR–RS+ channels show θ power that is lower than in SR+RS- or SR+RS+ channels and is not different from channels neither responding to the stimulus nor showing repetition suppression (SR–RS−). In the β range, SR–RS+ channels also show different β activity compared to SR +RS− channels. Additionally, we found very few sites with a negative stimulus response to the first stimulus in a series, suggesting that inhibition is a rare response to the stimuli.

Within the classical notion of RS, it is harder to explain the finding of RS without an initial SR. One possibility is that rather than simple adaptation, repetition suppression without stimulus response reflects direct top-down inhibitory prediction signals. This is consistent with the conjecture that top-down prediction effects are carried by activity in the beta band. Previous studies argued that feedforward and feedback signals are distributed across cortical layers and segregated by spectral content[27,28]. They suggested that feedback signals target deep (infragranular) layers of the cortex with activity in the β band. In addition to enhanced β band activity, repetition suppression sites showed enhanced β:HFA PAC. Hence, repetition suppression embedded in β activity might reflect an internal model based on stimulus history.

RS can be explained by neural sharpening[4,12,29,30] due to the fall-out of neurons that are not optimally tuned to stimulus features with repetition. Repeated sensory evidence strengthens intracortical inhibitory connections. This lateral interaction[31] may cause a decrease in the population response ('inhibitory sharpening'[12]). Even though we cannot directly test neural sharpening, lateral interactions can be an explanation for the SR +RS− vs SR–RS+ sites. This sharpening may also be influenced by top-down inhibition as a component of hierarchical predictive coding.

Predictive coding (PC) schemes mostly assume that the stimulus response itself is suppressed when it is predicted, indicating that the evaluation of a stimulus likelihood precedes or occurs simultaneously with the bottom-up response to the stimulus, so that only deviations from prediction are registered. The current results show that RS effects followed SR in time, overlapping in time with only the late part of the stimulus response. This is consistent with prediction-error-like effects as the mismatch negativity (MMN), which overlaps in time the late part of the auditory N1 response from 100 ms after stimulus onset[32]. Thus, even if predictions are formed a-priori, they seem to affect mainly the later stages of processing.

The SR+ and RS+ ensembles showed distinct spectra and spatial and temporal layout, yet were not disconnected—activity in the θ and β range exhibited significant phase-phase coupling and HFA showed information transfer from SR+ to RS+ sites, which was local in time. The phase coupling signals originate from separable sites and were found exclusively between θ and β, suggesting that the θ:β cross-frequency phase coupling effects are not due to waveform shape[33]. Phase coupling might enable temporally precise coordination of neuronal processing by establishing systematic spike-timing relationships among functionally distinct oscillatory assemblies enabling functional integration and coordination[34,35]. The functional meaning of this coupling in the present case remains to be investigated.

Another sign of communication between the sites is that HFA in the SR+ sites reduced the uncertainty of HFA responses in the RS+ sites. MI is based on information theory[36] and formally characterizes the information content of neural responses and interactions between these responses. In previous studies, MI has been applied to multiunit recordings and local field potentials in nonhuman primates[25,37–39] and intracranial data in human[40]. Importantly, MI makes no assumptions about the content of the signal itself but only that it changes as a function of time. The strength of this approach is that it characterizes the nonlinear relationship between two different neural responses. We found a clear exchange of information from SR+ to RS+ sites but not vice versa. Under the assumption that the repetition suppression reflects the process of top-down model predictions, this result supports the notion that stimulus responses (or prediction errors under the PC framework[41]) inform the generation of the internal model. Taking together the finding of phase:phase coupling between θ and β activity, the fact that each band respectively

modulated the local HFA amplitude in its region (PAC), and the finding of directional MI between of SR+ to RS+ sites, supports the proposal of cross-talk between bottom-up (θ) and top-down (β) effects.

**Limitations**. Repetition effects were investigated in depth in previous studies in vision and audition across different species using a large repertoire of recording techniques[19]. These studies are often motivated by behavioral repetition priming, showing that repetition leads to improved identification of stimuli[42]. Repetition-suppression is assumed to reflect statistical learning[43,44] and contributes to sensory memory update[45] by tracking stimulus history[46]. Most repetition suppression studies compared responses between a first and second presentation which show the strongest repetition suppression effects in non-invasive recordings[20,47–49]. Here, we calculated RS across the first three standards since we conjectured a monotonical decrease[50] until the fourth standard[45,47]. Several previous studies showed local PAC as in our study (phase of low frequency and amplitude of higher frequency of the same broadband signal) across species and different recording techniques[51–64]. However, since PAC supports information processing, genuine coupling in contrast to spurious correlations must be shown. Spurious coupling can result from filtering non-sinusoidal signal[34] creating artificial coupling at distinct frequencies, especially in local PAC. However, commensurate with phase:phase coupling, PAC is more likely genuine if the higher frequency is exclusively coupled to only one of two distinct low frequencies originating from separable neuronal processes. We indeed found a double dissociation of HFA SR+ coupled to θ but not β and vice versa for HFA RS+ in the temporal interval of coupling between θ and β. This differential coupling finding provides evidence for two distinct processes[34].

In our study, we focused on the repeated and hence predictable part of the auditory stimulation. How repetition and expectation suppression interact is still debated. Our paradigm does not allow us to conclusively dissociate between repetition independent of expectation. However, recent evidence[65] suggests that subjects are less likely to apply overt expectations (even when available) when the stimuli are not task-related (as was the case in our study), suggesting that when attention is directed elsewhere, expectation vanished despite robust repetition suppression being still evident[65,66].

## Conclusion
Our critical finding is that SR and RS dynamics were temporally and spatially distinct. Our results highlight the role of distinct processes in computing a stimulus response and feedback signals which are functionally linked but do not completely overlap.

## Methods
**Patients**. Ten patients (mean age 32, SD = 9.74) undergoing pre-surgical monitoring for drug-resistant epilepsy[67] with subdural electrodes participated in the experiment after providing their written informed consent. Recordings took place at the University of California, San Francisco (UCSF) (five patients) and at the Dept. of Epileptology (Krankenhaus Mara), Bielefeld University (five patients) and were approved by the local ethics committees. Data from these patients were preprocessed in an analogous manner as reported[18].

**Stimuli**. Participants listened to stimuli consisting of 180 ms long (10 ms rise and fall time) harmonic sounds with a fundamental frequency of 500 or 550 Hz and the 3 first harmonics with descending amplitudes (−6, −9, −12 dB relative to the fundamental). The stimuli were generated using Cool Edit 2000 software (Syntrillium, USA). They were presented from loudspeakers positioned at the foot of the subject's bed at a comfortable loudness.

**Procedure**. While reclined in their hospital bed, participants watched an engaging slide show while sound trains were played in the background. Sound trains included high probability standards ($p = 0.8$; $f_0 = 500$ Hz) mixed with low probability deviants ($p = 0.2$; $f_0 = 550$ Hz) in blocks of 400 sounds, with a stimulus onset asynchrony (SOA) of 600 ms. Hence, in each block, 320 standard tones and 80 deviant tones were presented. In different blocks, the order of the sounds was either pseudorandom, with a minimum of three standard tones before a deviant (irregular condition), or regular, such that the standard stimulus was repeated exactly four times before a deviant was presented (S-S-S-S-D-S-S-S-S-D-…, Fig. 1a). Thus, in the regular condition, the fourth standard tone was fully predictable, whereas in the irregular condition, the fourth stimulus could be either a standard or a deviant, and prediction was not possible. The current report examines the responses to the repeating standards. The deviance-related responses from the subset of the subjects recorded at UCSF was previously reported[18].

**Data recording and preprocessing**. ECoG was recorded at UCSF using electrode grids equipped with 64 platinum-iridium-electrodes, arranged in an $8 \times 8$ array with 10 mm center-to-center spacing (Ad-Tech Medical Instrument Corporation, Racine, Wisconsin). At The Mara, Bielefeld, ECoG was recorded via electrode strips (single strips or parallel arrangement of strips; white dots in Fig. 1b, d represent all electrode locations) using a Nihon Kohden amplifier (Tokyo, Japan). Electrodes were positioned based solely on clinical needs. The exposed electrode diameter was 2.3 mm. The data at UCSF were recorded continuously throughout the task at a sampling rate of 2003 Hz. At The Mara, the sampling rate was 2000 Hz in the case of four subjects and 1000 Hz in one subject. We used Matlab 2013b (Mathworks, Natick, USA) for all offline data processing. After visual inspection, we excluded channels exhibiting ictal activity or excessive noise from further analysis. In the remaining "good" channels (see Table 1), we then excluded time intervals containing artifactual signal distortions such as signal steps or pulses by visual inspection. Finally, we re-referenced the remaining electrode time series by subtracting the common average reference

$$x_{\text{CAR}}(t) = \frac{1}{n} \sum_{c=1}^{n} x_c(t) \tag{1}$$

calculated over the $n$ good channels from each channel time series $x_c$. The resulting time series were used to characterize brain dynamics of responses to repeated auditory stimulus presentation. For high-frequency signals, we band-pass filtered each electrode's time series in the high-frequency range (80–150 Hz). All filtering was done with zero-shift infinite impulse response (IIR) filters [Butterworth filter of order 4: filtfilt() function in matlab]. We obtained HFA by calculating the analytic amplitude $A_f(t)$ by Hilbert-transforming the filtered time series. We smoothed the HFA amplitude time series such that the amplitude value at each time point $t$ is the mean of 10 ms around each time point $t$. Filtering was done for each trial (-1 s to 2 s around stimulus onset—sufficiently long to prevent any edge effects during filtering).

**Data analysis**. We conducted the following analysis steps explained in detail below. First, we defined stimulus response of the HFA (I-Stimulus-responsive activity modulation). Next, we parameterized response attenuation of cortical HFA responses to repeated standard tones using a time-resolved ANOVA (II – Repetition-suppression). We then compared the temporal and spatial profile of stimulus response and repetition suppression (III – Comparison of stimulus response and repetition suppression). Next, we tested to what degree the HFA SR and RS are associated with low-frequency activity (IV – Comparison of low-frequency specific modulation). We then tested whether HFA SR and RS were modulated by distinct neural populations in low frequencies (V – Cross-frequency modulation) and for functional integration between low-frequency neural populations (VI – SR:RS integration). Finally, we assessed information flow between SR + and RS + channels using time-resolved mutual information to test for the directionality of information propagation (VII – Information propagation).

**I – Stimulus-responsive activity modulation**. We identified stimulus-responsive channels SR+ showing a significant HFA modulation following the onset of standard stimuli using the following steps. We first averaged stimulus-locked HFA responses across all standard trials. To apply a sensitive measure which takes into account that HFA responses can occur with a delay and/or can be transient, we calculated $\bar{A}_{\text{HFA}}$ as averaged activity modulation across five different intervals, each with a duration of 100 ms following the stimulus onset (starting at 0, 50, 100, 150, and 200 ms). This allowed us to select fast and delayed responding channels. We then calculated the average baseline activity $\bar{B}_{\text{HFA}}$ across the 100 ms preceding the stimulus onset. For the stimulus-related HFA, we subtracted baseline $\bar{B}_{\text{HFA}}$ from the activity modulation $\bar{A}_{\text{HFA}}$ following stimulus onset for each of the five temporal intervals. The difference between $\bar{B}_{\text{HFA}}$ and $\bar{A}_{\text{HFA}}$ was compared against a surrogate distribution derived from randomly shifted time series (1000 permutations). In each iteration, the time series of each channel and each trial were shifted (circular shift of the entire trial time series between −0.1 and 0.6 s separately, and new (surrogate) trial averages ($\bar{B}_{\text{surr}}$ and $\bar{A}_{\text{surr}}$) were calculated from the shifted trials, resulting in a surrogate distribution of differences between baseline and activity. The comparison of observed difference values with the surrogate distribution results in a $p$ value for each channel and time interval. To control for multiple comparisons, we corrected the $p$ values by applying the false discovery rate (FDR)[68] method across all channels and time intervals. Channels with a $q < 0.05$ (where $q$ is

the false discovery rate) in any of the five intervals were classified as showing a significant HFA modulation following the standard stimuli and were denoted as SR+, whereas the remaining channels were labeled as SR-. To determine the amount of evidence for a change over baseline, we compared HFA values at each time point with HFA values in the baseline interval separately for each channel across trials, using Bayes factor (BF; bf.m toolbox in MATLAB http://klabhub. github.io/bayesFactor/). BF >3 is considered strong evidence for a difference (the difference is three times more likely than no difference) and BF <1/3 supports null effects[69,70].

*II – Repetition-suppression.* RS is defined as attenuated amplitude to repeated stimulus presentation. Hence, this definition is twofold: (i) a change of amplitude which is (ii) monotonically decreasing with the number of repeated stimulus presentations. While (i) refers to statistical differences in brain response with repetitions, (ii) assumes a specific model of response attenuation. We thus grouped trials according to the number of standards in a train in three groups ($S_1$, $S_2$, and $S_3$) since only the first three standards in a train can be expected in both conditions. To parameterize the amplitude modulation with stimulus repetition (i), we ran a one-way ANOVA with a factor number of standards for each electrode (with trials as a random variable), regardless of whether it was SR+ or SR−, at every time point, both in the regular and irregular condition. This leads to an $F$ value time series (main effect: $F_{N\ standard}$) for each channel in each condition. Significant $F$ values only define differences between numbers of preceding standards but not the exact model of monotonical decrease of neural responses. We tested (ii) the model of a monotonic neural amplitude decrease across the number of repetitions. For each electrode, at each time point, we calculated the Pearson correlation between the mean HFA and the number of sequential preceding standards, yielding a time series of correlation coefficients ($r$) for each channel. We compared each $F$ value and $r$ value against a surrogate distribution constructed under the null assumption of no difference or no correlation, respectively. This surrogate distribution was constructed by randomly reassigning the labels ($S_1$, $S_2$, and $S_3$) to the single trials in 1000 permutations for each channel. This leads to 1000 surrogate $F_{Nstandard}$ and $r$ value time series. We assigned a $p$ value to each $F$ and $r$ value within the surrogate distributions. The $p$ values for $F$ and $r$ were corrected for multiple testing by applying the FDR. $F$ and $r$ with $q < 0.05$ were classified as significant. Finally, channels showing intervals with the conjunction of both significant $F_{N\ standard}$ and significant negative $r$ value were considered as showing significant RS and were labeled RS+. In contrast, channels with temporal intervals of significant $F$ value and a significant positive $r$ value show repetition enhancement (RE+). In addition, to ensure that RS+ shows an $S_1 > S_2 > S_3$ relationship, we computed the amplitude decrease over a train of tones across these channels. We averaged mean responses following $S_1$, $S_2$, and $S_3$ across RS+ channels. We then calculated the differences between those responses (response following S3 vs response following $S_2$ and response following $S_2$ vs response following $S_1$, respectively) at each time point and summed the sign function of differences (−1 and +1 for negative and positive differences, respectively).

$$DiffSR = sgn(S_1 - S_2) + sgn(S_2 - S_3) \quad (2)$$

Positive values of DiffSR indicate a $S_1 > S_2 > S_3$ relationship. Higher positive values of DiffSR indicate stronger differences between S1, S2, and S3.

*III – Comparison of stimulus response and repetition suppression.* We tested the spatial and temporal relation between stimulus response (channels selected in step I) and repetition suppression (channels selected in step II) in the following way. If SR and RS are multiplicatively related[21], channels with stronger stimulus response should show stronger repetition suppression. Alternatively, suppression could be subtractive (the same amplitude reduction regardless of SR), in which case RS will be fixed, that is, not dependent on SR. To examine this, we tested the correlation between SR and RS measures across channels. For each channel, we averaged HFA in response to $S_1$ in the time window of strongest SR, averaged $F_{N\ standard}$ ($\bar{F}$) and $r$ ($\bar{r}$) values separately (section II) in the interval of significant RS, and separately correlated the $S_1$ HFA with $\bar{F}$ and with $\bar{r}$. This correlation was tested both across all channels and across channels showing a significant stimulus response (as defined in I above) and/or a significant RS+ (as defined in II above).

*IV – Comparison of dominant band power.* We estimated the power spectral density (PSD) in each trial (collapsing across all $S_1$, $S_2$, and $S_3$ trials) separately for SR+ and RS+ channels using Welch's method based on the FFT[71]. Specifically, for each channel, we calculated PSD as a function of frequency (1–30 Hz, 1 Hz steps) in each trial in temporal intervals of 100 ms (due to the short SOA of 600 ms) between −0.1 s to 0.3 s in steps of 50 ms. The resulting PSD values were averaged across trials yielding one PSD for each channel. We then averaged across three canonical low-frequency bands θ (4–8 Hz), α (8–12 Hz), and β (15–30 Hz) and compared the resulting power estimates across channels using a two-way ANOVA with the factors channel type (SR+RS−, SR−RS+, SR−RS−, and SR+RS+) and frequency band (θ, α, β). Post hoc we compared the power estimates between channel sets, separately for the three different frequency bands by computing $t$ values and comparing those to the critical $t_{crit}$. $t_{crit}$ denotes the critical $t$ value which the observed $t$ values had to exceed to be considered significant. To correct

for multiple comparisons, $p$ values were assigned to each $t$ value within a surrogate distribution constructed by randomly assigning labels (SR+RS−, SR−RS+, SR−RS−, and SR+RS+ individual channels) and corrected by applying the FDR. The adjusted $p$ values are labeled $q$. T values with a corresponding $q < 0.05$ (corrected $p$ value) were classified as statistically significant.

*V – Cross-frequency coupling.* The interplay between activity at distinct frequencies is proposed to be regulated via cross-frequency phase coupling (CFPC[72]) and via phase–amplitude cross-frequency coupling (PAC; see below[73–75]. Phase–amplitude cross-frequency coupling (PAC) is a mechanism that has been proposed to coordinate the timing of neuronal firing within local neural networks (see ref. [73] for a review). We utilized conventional cross-frequency coupling metrics[51,76] to test for differences in coupling of HFA to low-frequency bands in SR+ vs. RS+ channels. We calculated the instantaneous phase for low-frequency activity (see below) for each SR+ and RS+ channel time series. In the temporal interval of coupling between low-frequency networks as determined in the previous step IV, we divided both the θ and β cycle separately in 50 equally spaced bins ranging from −π to π and computed the average HFA for all trials within a 45-degree window centered on every phase bin[25]. The resulting HFA histograms—each containing 50 values— were averaged separately for SR+ and RS+, separately for θ and β activity. We then calculated the normalized Kullback–Leibler divergence (KLD) of the observed distribution against a uniform distribution to quantify how strongly the observed distribution of HFA SR+ and HFA RS+ were modulated by the phase of the θ or β activity. The obtained KLD were compared in a two-way ANOVA with the factor channel type (RS+ vs. SR) and frequency band (θ vs. β). The interaction effect of the ANOVA describes the double dissociation of HFA of SR+ and RS+ PAC to θ and β networks, respectively.

*VI – SR:RS integration.* We hypothesized that while the amplitude of HFA is modulated by the phase of the low frequencies within a population (PAC[14]), communication across populations in low frequencies might be achieved via CFPC[34,52,77]. CFPC is defined by a nonrandom phase difference between oscillations, enabling temporally precise coordination or integration among functionally distinct oscillatory networks[34]. We tested whether phases of canonical low-frequency bands $f_1$ and $f_2$—either θ:α, α:β, or θ:β—are aligned and whether this interaction is modulated in time. To that end, we calculated the instantaneous phase of SR+ and RS+ channel time series and binned phase time series in intervals of one cycle of the $f_1$ (133 ms for θ and 100 ms for α) centered on time points between −200 to 300 ms following stimulus onset in each trial. In each temporal interval in each trial, we divided the $f_1$ cycle into 50 equally spaced bins ranging from −π to π and registered the $f_2$ phase at each bin. This was done for each pair of SR+ and RS+ channels, excluding SR+RS+ channels within each single recording session and separately for each patient, in which we found both RS and SR channels. This results in an $f_2$ phase angle distribution at each $f_1$ phase and at each time point for each pair. For each distribution, we calculated the concentration coefficient κ (reciprocal value to variance)

$$\kappa = \frac{1}{\sigma^2} \quad (3)$$

across all $f_2$ phase angles at each $f_1$ phase at each time point. κ time series of each SR+/RS+ channel pair were baseline corrected by subtracting the mean κ values in the 200 ms preceding stimulus onset. We then averaged κ values across all f1 phases leading to a κ time series for each pair of SR+/RS+ channels. Temporal intervals of high κ (low variance of $f_2$ phases) indicate coupling of the $f_2$ phase to $f_1$ phases. Low κ, on the other hand, indicates that $f_2$ and $f_1$ phases are unrelated. Each κ-value was compared against a surrogate distribution. In 1000 runs, we shifted phase time series in each trial and each channel separately and calculated surrogate κ values. To correct for multiple comparisons, $p$ values were assigned to each κ-value within the surrogate distribution and corrected by applying the FDR procedure. We then compared $\kappa_{\theta:\beta}$ with $\kappa_{\theta:\alpha}$ and $\kappa_{\alpha:\beta}$ time series. To parameterize the difference in $f_1{:}f_2$ coupling, we ran a one-way ANOVA with factor frequency pairs (θ:β, θ:α, and α:β) at each time point. This leads to an $F$ value time series representing the difference in coupling strength between θ, α, and β pairs. Each $F$ value was compared against a surrogate distribution and a p value within the surrogate distribution was assigned. This surrogate distribution was constructed by randomly reassigning the labels (θ:β, θ:α, and α:β) to the f1:f2 time series in 1000 permutations leading to 1000 surrogate $F$ values. To correct for multiple comparisons, we corrected the $p$ values resulting from comparing $F$ values against the surrogate distribution by applying the FDR.

*VII – Information propagation.* Using mutual information (MI), we tested whether there is directed information flow between SR+ and RS+ channels. MI measures how much a random variable can be predicted by another random variable. Specifically, MI quantifies the uncertainty about one random variable given knowledge of another variable and is given in units of bits. Mathematically MI was calculated as the sum of entropies of SR+ and RS+ minus their joint entropy. In each subject, we calculated MI between all pairs of RS+ and SR+ channels, excluding SR+RS+ channels within each subject. This was done using 200 ms intervals around each time point of RS with 200 ms intervals around each time point of SR+ time series

between −0.1 and 0.4 s for trial-averaged HFA.

$$MI = \frac{H(SR) + H(RS) - H(RS, SR)}{-\log 2\left(\frac{1}{200ms}\right)} \quad (4)$$

Where H(SR) and H(RS) stands for the entropy of the SR+RS− and the SR−RS+ channel, respectively, and H(RS, SR) designates their joint entropy. The denominator standardizes each value by the maximal achievable information value. We iterated through all intervals around each time point of RS+ and SR+ channels resulting in a matrix of MI values quantifying which temporal interval of the HFA time series of SR+ channels predicts the HFA time series of RS+ channels and vice versa for each pair of channels. We then averaged the MI across all pairs of channels. We then compared each MI-value against a surrogate distribution. This surrogate distribution was constructed by randomly shifting SR+ and RS+ time series of single channels and averaging across subjects in 1000 permutations. In each run, we repeated the same analysis as outlined above. This leads to 1000 surrogate MI values. The resulting $p$ values for the MI values relative to the surrogate distribution were corrected by applying the FDR.

**Reporting summary**. Further information on research design is available in the Nature Research Reporting Summary linked to this article.

## Data availability
The datasets generated and/or analyzed during the current study are available in the Open Science Foundation repository (https://osf.io/ceutw/?view_only=1d6ba767ebe3458ca26ade4945972d8c).

## Code availability
Custom MATLAB 2013b code used for preprocessing and analysis is available as a GitHub repository (https://github.com/repetitionsuppression/-ECOGRepetition Suppression), which includes system requirements and dependencies.

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

## Author contributions

S.D., L.Y.D., and R.T.K. conceived and designed the experiment. S.D. collected the ECOG data. C.G.B. provided clinical information. S.D., D.E., and C.R. analyzed the data, S.D., D.E., H.-J.H, L.Y.D., and R.T.K. interpreted the data S.D., D.E., L.Y.D., and R.T.K. wrote the manuscript.

## Funding

## Competing interests

The authors declare no competing interests.
