## [Peer Review File · Communications Biology]

Reviewers' comments:

Reviewer #1 (Remarks to the Author):

This paper examines neural responses in an auditory oddball paradigm in human epileptic patients. They analyzed field potentials to 3 repeated tones and find that although many recording channels were responsive to the tones only a small proportion of these showed a suppressed high gamma-band response to the repeated tones ("repetition suppression"). Another small proportion of channels did not surpass their statistical criterion of a response to the first stimulus of a sequence but nonetheless showed repetition suppression. They report some additional findings regarding interactions between different frequency bands that differed between the responsive and unresponsive channels. They focus their report on comparing the responsive channels and the unresponsive ones without repetition suppression, arguing for a dissociation between the response to the first stimulus of a sequence and repetition suppression. However, as I explain below, their evidence for such a dissociation is rather weak.

Specific comments.

1. The authors performed several analyses on the data of this interesting data set of invasive recordings in human patients. However, I am not convinced by their results. The statistical evidence for the main finding they focus on, i.e. the presence of repetition suppression in unresponsive channels, is weak. First, their mean Bayes factor for the null hypothesis, i.e. no response in the channel, is low: $1/0.41 = 2.4$, which is according to common standards only "anecdotal" evidence for the null hypothesis. Second, their Figure 1h appears to show a response to the first stimulus in the "unresponsive" channels: the dark blue curve is above baseline after stimulus onset. Third, they should provide the Bayes factor for the repetition suppression effects so that one can assess the evidence for these effects in the unresponsive (and responsive) channels.

2. On a more conceptual level, they should consider that they analyze a population response, i.e. the high gamma-band power, which reflects the response of a large number of neurons. It is well possible that the "unresponsive" channels that show repetition suppression reflect the summed response of neurons excited by the stimulus, which show repetition suppression, and of neurons that are inhibited by the stimulus. For such a population of heterogeneous neurons, the summed activity will be close to baseline but there will be measurable repetition suppression due to decreased response by the excited neurons. Thus, it is difficult if not impossible to make claims about dissociation between repetition suppression and response using such a (relatively) low spatial resolution measure (relative to single neurons) as high gamma-band activity.

3. The paradigm employed by the authors confounds adaptation/repetition suppression and expectation suppression which are empirically distinct phenomena, although both might in principle be explainable by predictive coding. The problem is that the repeated stimuli are both predictable and repeated: how can one distinguish an effect of repetition (i.e. repetition suppression/adaptation) from an effect of prediction/expectation? Given this issue, the discussion of their results in terms of predictive coding is highly speculative.

4. The analyses of Figure 2 are interesting but one cannot know from these analyses whether the differences are because of response versus no (or weak) response to the stimulus or repetition suppression versus no repetition suppression since these two factors are confounded/correlated in their analysis: the authors compare responsive channels without repetition suppression with unresponsive channels with repetition suppression. I do not see the purpose of these analyses in the context of the present paper.

5. It is close to impossible to fully understand the analyses or Figures without reading the Methods which is presented as the Supplem. Text. This should be remedied. After reading the Suppl. Text on Methods, details of some Figure panels were still unclear. (a) Are the bands/shading standard errors of the mean or confidence intervals? (b) The F values that are shown in Figure 1 and Figure 2 cannot be F values from an ANOVA since such F values cannot be negative. Also, if F values are close to 0 then typically the model assumptions are not met. Are the shown "F values" baseline corrected (which does not make sense to me)? What does the stippled horizontal line correspond to? The latter also holds for

other panels of Figures 1 and 2 where horizontal lines are presented but left unexplained. (c) What is diffSR in Fig 1h?

6. Also in the text, clarifications are needed: line 76: what does the Greek symbol indicate? Line 83: what is tcrit ?

7. Methods: (a) Why did the authors not use a sign test or Wilcoxon signed-rank test to test for significant responses (paired, trialwise comparison of baseline and stimulus period)? The method they employ now by shifting in a circular way trial times seems to be needlessly complicated. (b) how was the FDR applied? Taking into account the number of channels and number of time points? Please clarify? (c) Did the authors observe repetition enhancement, i.e. positive r between trial number and response? (d) How did they compute the Bayes factor?

Reviewer #2 (Remarks to the Author):

Ten pre-surgery patients monitored with subdural electrodes heard a long sequence of sounds while watching a slideshow. In blocks of 400 sounds 80% were sound A and 20% were sound B, and within each block sequences were either pseudorandom or a regular train of four As followed by a B. Recordings of high frequency activity were characterised as stimulus response (SR, indicating stimulus evoked amplitude modulation of the response) or repetition suppression (RS, indicating a reduction in the amplitude between successive repeats of a tone) on each recording channel. Of 412 recording sites, 115 responded to the auditory stimulus. Of these 84 were classed as SR but not RS and 24 were classed as RS but not SR. The remaining 7 were classed as both SR and RS. The analysis supports a general dissociation of these two response characteristics between recording sites and between frequency bands. It also suggests differences in the timing of the SR and RS components of the response in sites classed as both SR (for which stimulus modulation was evident in the earlier component) and RS (for which attenuation was evident in the later component). The authors consider these dissociations as evidence that challenges the predictive coding framework.

Overall, I found the study to report a thorough analysis of an interesting and important dataset carried out to a high standard. I consider the conclusions to be supported by the results/analysis. And the implications of the data in terms of the predictive coding framework make this paper a welcome contribution that will help advance the understanding of this important topic, and that of repetition suppression more generally.

However the manuscript is unnecessarily difficult to read in its present format. It is very short, and begins with a results section without identifying/explaining the research question. I had to read the supplementary materials carefully before I could really understand the main text, not least due to the dense packing of abbreviated terms, several of which were undefined (see minor comments). I therefore recommend the following revisions.

1. Include an introduction section that identifies the specific research question in terms of testing the predictions/assumptions of the predictive coding theory, and which motivates the specific design of the study as an appropriate way of carrying out that investigation. This section could also emphasise the novelty of this dataset (human electrode recordings) in the context of other primate single-unit recording studies of repetition suppression (e.g., Li L, Miller EK, Desimone R. The representation of stimulus familiarity in anterior inferior temporal cortex. *Journal of Neurophysiology*. 1993; 69(6):1918–1929.)

2. Incorporate the bulk of the supplement (particularly the analysis subsections) into a methods section in the main text so that the manuscript explains the various quantities that are subjected to analysis prior to presenting the results of those analyses.

3. Improve the clarity of the figures.

3a. Figure 1: The authors should add a descriptive title that summarises the take-home point of the figure. The text in the figure is unreadable in places and should be enlarged throughout. Presumably the top and bottom traces in a depict the regular and pseudorandom blocks (please clarify with a label)? Please state what $\text{Diff}_{\{\text{SR}\}}$ means in the caption for panel h and also label the righthand y-axis. Consider adding labels SR+RS- and SR-RS+ to panels i and j. I would recommend unpacking some of the abbreviations in the caption to make it more readable.

3b. Figure 2: The authors should add a descriptive title that summarises the take-home point of the figure, and increase the size of the text throughout. Some of the labels, e.g., 'phase:phase frequency' are difficult to understand given that they are explained in the supplement rather than the main text.

Minor comments:

4. Line 50 - SR+RS+ — it would help clarify that this is the authors' label for the condition where RS and SR overlap (e.g., use 'designated' as in later examples).

5. Line 54 - S_1-S_3 — on first reading this appears to be a subtraction of two quantities, rather than a reference to three repeats of a given stimulus sound, please clarify.

6. Line 56 - S1 should be S_1 and S3 should be S_3 ? Also RS should be described as a significant reduction of activity from S_1 to S_2 and from S_2 to S_3 , to reflect the calculations described in the supplement.

7. Line 52 and Supplement line 31 - please define ECoG. I assume you mean electrocorticography but I doubt the general reader will be familiar.

8. Supplement line 49 - please define HFA. It is defined only in the abstract.

9. Supplement line 87 - monotonously should be monotonically?

10. Supplement line 105 - please define 'the FDR procedure'

11. Supplement line 93 - please give a definition for SR- in the supplement.

12. Supplement line 147 - closing bracket missing.

13. Supplement line 156 - should RS be RS+ here?

14. Supplement line 160 - 'parameterizes' should instead be e.g., 'measures' or 'describes' etc.

15. Finally, the authors may wish to take a look at a recent paper of mine in which it is suggested that repetition suppression may reflect the sharpening of neuronal receptive fields via plasticity mediated by recurrent inhibitory interactions. Please feel under no obligation at all to cite this paper - I mention it because the reported temporal dissociation of SR and RS at the SR+RS+ sites seems to me to be consistent with a role of lateral inhibition in attenuating the neuronal responses to repeated stimuli: Spigler G, Wilson SP (2017) Familiarization: A theory of repetition suppression predicts interference between overlapping cortical representations. PLoS ONE 12(6): e0179306. doi:10.1371/journal.pone.0179306

Stuart Wilson

Reviewer #3 (Remarks to the Author):

In this manuscript, the authors studied stimulus-response and repetition-suppression with ECoG. Their main finding is that stimulus-response and repetition-suppression dynamics revealed temporally and spatially dissociated but interacting neuronal activity. This is a highly novel and exciting finding in the field. The discovery will undoubtedly form the basis for several future studies and enlighten the previous results on repetition-suppression. The conclusions of the manuscript are original. Overall, the manuscript is well-written, logical, and scientifically rigorous. I only have a little comment. The authors should define the abbreviation PC.

Dear Editor,

We are grateful for the encouraging reviews and for the helpful suggestions made by the three referees, which allowed us to improve the manuscript. Below please find our response to each of the reviewers' comments

Reviewers' comments:

Reviewer #1 (Remarks to the Author):

This paper examines neural responses in an auditory oddball paradigm in human epileptic patients. They analyzed field potentials to 3 repeated tones and find that although many recording channels were responsive to the tones only a small proportion of these showed a suppressed high gamma-band response to the repeated tones (“repetition suppression”). Another small proportion of channels did not surpass their statistical criterion of a response to the first stimulus of a sequence but nonetheless showed repetition suppression. They report some additional findings regarding interactions between different frequency bands that differed between the responsive and unresponsive channels. They focus their report on comparing the responsive channels and the unresponsive ones without repetition suppression, arguing for a dissociation between the response to the first stimulus of a sequence and repetition suppression. However, as I explain below, their evidence for such a dissociation is rather weak.

Specific comments

1. The authors performed several analyses on the data of this interesting data set of invasive recordings in human patients. However, I am not convinced by their results. The statistical evidence for the main finding they focus on, i.e. the presence of repetition suppression in unresponsive channels, is weak. First, their mean Bayes factor for the null hypothesis, i.e. no response in the channel, is low: $1/0.41 = 2.4$, which is according to common standards only “anecdotal” evidence for the null hypothesis.

Reply: We agree with the reviewer that a prerequisite to delineate the double dissociation between SR and RS is to show that SR–RS+ channels do not show SR and that SR+RS– channels do not show RS, which puts a lot of weight on supporting null effects. We did this in the previous version in two ways. First, we defined stimulus response (SR) as the HFA exceeding baseline activity. A critical aspect here is that these channels have to exceed a conservative confidence interval of baseline activity corrected for multiple comparisons. We labeled these channels SR+. To make sure that SR– channels indeed had no systematic response to the stimulus their activity had to remain below even the liberal uncorrected significance threshold. Indeed, we reported channels with significant RS which do not show even a trend toward significance. The respective part in the results section reads “Channels designated as SR-RS+ showed no trend towards a stimulus response (average $p = 0.41$, $SD = .1$) even with a liberal, uncorrected significance criteria.”. Second, we used Bayes Factors (BF) to determine the evidence for no stimulus response in these channels. In our previous analysis we estimated BF across channels. The BF across channels indeed showed more support for no response than to response, but only weakly so ($BF = 0.41$). We attribute this weak statistical support in part to the fact that BF depends non-linearly on sample size and we have only a limited number of channels (i.e. $N_{SR-RS+} = 24$). The reviewer found the BF not convincing enough, and we appreciate the request for stronger statistical evidence. To address this issue in a new analysis we assessed the evidence for a stimulus response separately in each channel across trials. In each channel, we compared baseline HFA values with HFA values at each time point. The resulting BF time series for each of the 412 channels were grouped in the channel sets SR+RS–, SR–RS+, SR+RS+, and SR-RS–, based on the corrected and uncorrected permutation thresholds described above. We found that average BF values in SR– channels (SR–RS+ and SR–RS– channels) do not exceed $BF = .3$. Moreover, the maximal BF value in SR–RS+ channels is .19. That is, absence of response was at least 5.26 times more likely than presence of response, corresponding to strong evidence for the null hypothesis. Extreme evidence for a stimulus response was observed only in SR+ channels

(SR+RS–: $BF_{\text{mean}} = 54.54$, $BF_{\text{min}} = 34.5$, $BF_{\text{max}} = 506.19$, and SR+RS+: $BF_{\text{mean}} = 570.03$, $BF_{\text{min}} = 289.7$, $BF_{\text{max}} = 7590$).

In the new version of the manuscript, we only report the BF estimation in each channel. The respective paragraph in the Methods section now reads:

“To determine the amount of evidence for a change over baseline we compared HFA values at each time point with HFA values in the baseline interval separately for each channel across trials, using Bayes factor (BF; bf.m toolbox in MATLAB <https://klabhub.github.io/bayesFactor/>). $BF > 3$ is considered strong evidence for a difference (difference is 3 times more likely than no difference) and $BF < 1/3$ supports null effects [22, 23]”

The respective paragraph in the Results section now reads:

“We also estimated the BF to determine the amount of evidence for a change over baseline separately for each channel at each time point. Bayes factor analysis provided strong support to lack of stimulus response in SR–RS+ channels ($BF_{\text{mean}} = .12$, $BF_{\text{min}} = .098$, $BF_{\text{max}} = .19$; **Fig 2c**), and SR–RS– channels ($BF_{\text{mean}} = .17$, $BF_{\text{min}} = .15$, $BF_{\text{max}} = .21$), while strong evidence for stimulus response was observed in SR+RS+ channels ($BF_{\text{mean}} = 570.03$, $BF_{\text{min}} = 289.7$, $BF_{\text{max}} = 7590$), and in SR+RS– channels ($BF_{\text{mean}} = 54.54$, $BF_{\text{min}} = 34.5$, $BF_{\text{max}} = 506.19$). We additionally estimated the BF to determine the amount of evidence for repetition suppression separately for each channel. We estimated the RS effect in each S1,S2,S3 sequence as the summed difference between HFA to S1,S2,S3 (response following S3 vs response following S2 and response following S2 vs response following S1, respectively). This gives a difference wave for each sequence presented. We found support for no repetition suppression, neither in SR+RS– ($BF_{\text{mean}} = .23$, $BF_{\text{min}} = .21$, $BF_{\text{max}} = .26$) nor SR-RS- channels ($BF_{\text{mean}} = .23$, $BF_{\text{min}} = .22$, $BF_{\text{max}} = .25$). However, we observed positive evidence in SR+RS+ channels ($BF_{\text{mean}} = 315.72$, $BF_{\text{min}} = 172.7$, $BF_{\text{max}} = 1928.4$) and in SR–RS+ channels ($BF_{\text{mean}} = 16.52$, $BF_{\text{min}} = 4.49$, $BF_{\text{max}} = 376$; **Fig 2d**).”

We therefore exchanged former fig2c and d with the current figure 2c, figure 2 is now:

The caption for Fig2 now reads:

“Figure 2: stimulus response and repetition suppression show different temporal profiles. a,b show the HFA amplitude modulation of SR+RS- (a) and SR-RS+ (b) channels to S₁, S₂ and S₃, the dashed black curve in (b) represents the summed sign of differences between S₁ & S₂ and S₂ & S₃ (Diff_{SR}). Positive values of this summed differences indicate a S₁>S₂>S₃ relationship, shaded areas represent the standard error of the mean. c, d show time-resolved BF for modulation of HFA over baseline (evidence for stimulus response) averaged separately across the four different channel sets (SR-RS+, SR+RS-, SR-RS-, and SR+RS+) (c) and for the effect of repetition suppression averaged across channels, shaded areas represent the standard error of the mean (d). Only SR+RS- and SR+RS+ channels show evidence for a stimulus response, only SR-RS+ and SR+RS+ channels show evidence for RS. e shows peak

latencies for RS (F-values, blue) and SR (HFA, green), compared within SR+RS+ channels (upper bars) and between SR+RS- and SR-RS+ channels (lower bars), error bars represent the standard error of the mean.”

Second, their Figure 1h appears to show a response to the first stimulus in the “unresponsive” channels: the dark blue curve is above baseline after stimulus onset.

Reply: Figure 1h (which in the current version is Fig 2b) shows two things. The dark blue curve has an upward trend and in the baseline interval the standard error around the mean is low. While indeed there is a nominal elevation of HFA, our previous analyses, and even more so the revised analysis reported above, shows strong support for the null hypothesis in SR- channels ($BF_{\text{mean}} = .17$, $BF_{\text{max}} = .21$). This is demonstrated by the following figure showing a plot of each time course for all single electrodes.

Figure. HFA amplitude modulation of SR-RS+ channels following stimulus presentation.

Third, they should provide the Bayes factor for the repetition suppression effects so that one can assess the evidence for these effects in the unresponsive (and responsive) channels.

Reply: We agree with the reviewer that this is helpful for the readers to assess the effects. As suggested by the reviewer, we repeated the Bayes analysis for the repetition suppression effect. We estimated the RS effect in each sequence (a sequence comprises all standards S_1, \dots, S_N in a row followed by a deviant D) as the summed difference between HFA to S_1, S_2, S_3 (response following S_3 vs response following S_2 and response following S_2 vs response following S_1 , respectively), considering the decrease from one standard to its subsequent standard (i.e., repeated stimulus) each instead of a vague global difference S_1-S_3 . This gives a difference wave for each sequence presented. We found no evidence for repetition suppression neither in SR+RS- ($BF_{\text{mean}} = .23$, $BF_{\text{min}} = .21$, $BF_{\text{max}} = .26$) nor SR-RS- channels ($BF_{\text{mean}} = .23$, $BF_{\text{min}} = .22$, $BF_{\text{max}} = .25$), however, positive evidence in SR+RS+ channels ($BF_{\text{mean}} = 315.72$, $BF_{\text{min}} = 172.7$, $BF_{\text{max}} = 1928.4$) and in SR-RS+ channels ($BF_{\text{mean}} = 16.52$, $BF_{\text{min}} = 4.49$, $BF_{\text{max}} = 376$). We added the new results to the results section in the new manuscript. The corresponding paragraph now reads:

“We additionally estimated the BF to determine the amount of evidence for repetition suppression separately for each channel. We estimated the RS effect in each S_1, S_2, S_3 sequence as the summed difference between HFA to S_1, S_2, S_3 (response following S_3 vs response following S_2 and response following S_2 vs response following S_1 , respectively). This gives a difference wave for each sequence presented. We found support for no repetition suppression, neither in SR+RS- ($BF_{\text{mean}} = .23$, $BF_{\text{min}} = .21$, $BF_{\text{max}} = .26$) nor SR-RS- channels ($BF_{\text{mean}} = .23$, $BF_{\text{min}} = .22$, $BF_{\text{max}} = .25$). However, we observed

positive evidence in SR+RS+ channels ($BF_{\text{mean}} = 315.72$, $BF_{\text{min}} = 172.7$, $BF_{\text{max}} = 1928.4$) and in SR-RS+ channels ($BF_{\text{mean}} = 16.52$, $BF_{\text{min}} = 4.49$, $BF_{\text{max}} = 376$; **Fig 2d**.)”

2. On a more conceptual level, they should consider that they analyze a population response, i.e. the high gamma-band power, which reflects the response of a large number of neurons. It is well possible that the “unresponsive” channels that show repetition suppression reflect the summed response of neurons excited by the stimulus, which show repetition suppression, and of neurons that are inhibited by the stimulus. For such a population of heterogeneous neurons, the summed activity will be close to baseline but there will be measurable repetition suppression due to decreased response by the excited neurons. Thus, it is difficult if not impossible to make claims about dissociation between repetition suppression and response using such a (relatively) low spatial resolution measure (relative to single neurons) as high gamma-band activity.

Reply: We thank the reviewer for this insight about an alternative interpretation of SR-RS+ channels according to which these channels reflect the sum of genuine SR+RS+ channels and neurons which are inhibited by the stimulus (and presumably have no repetition effects) and thus cancel out the overall stimulus response. This is a valid option to consider in these mesoscopic recordings. However, our results do not support this option. If apparent SR-RS+ channels represent a composite signal of genuine SR+RS+ excitatory and inhibitory neurons, then SR-RS+ channels should show comparable spectral signatures as other SR+ channels (namely SR+RS- and SR+RS+). However, both in the θ and the β range SR-RS+ channels show a spectral signature which resembles other SR- channels (SR-RS-) and differs from SR+ channels. Specifically, in the θ range SR-RS+ channels show θ power that is lower than in SR+RS- or SR+RS+ channels and is not different from channels neither responding to the stimulus nor showing repetition suppression (SR-RS- see comment 4). In the β range, SR-RS+ channels show different β activity compared to SR+RS- channels. Hence, critically, also SR-RS- show lower β values than SR-RS+ channels (see second next comment). Those distinct spectrum signatures belonging to distinct channel sets strongly support the hypothesis of dissociations in generators of repetition suppression and stimulus response

We added a paragraph to the discussion which now reads:

“An alternative explanation suggested by a reviewer is that SR-RS+ channels reflect the summed response of neurons excited by the stimulus and showing repetition suppression (i.e. genuine SR+RS+), and of neurons that are a-priori inhibited by the stimulus. These inhibited neurons would ‘cancel out’ the apparent SR in our mesoscopic recording of LFPs, making it look as if RS is present without SR. Without extensive single neuron measurements it is difficult to rule this option out. However, we suggest that this option is less likely, considering the different spectral characteristics of SR+ sites and RS+ sites. If apparent SR-RS+ channels represent a composite signal of genuine SR+RS+ and inhibitory neurons, then SR-RS+ channels should show comparable spectral signatures as other SR+ channels (namely SR+RS- and SR+RS+). However, both in the θ and the β range SR-RS+ channels show a spectral signature which resembles other SR- channels (SR-RS-) and differs from SR+ channels. Specifically, in the θ range SR-RS+ channels show θ power that is lower than in SR+RS- or SR+RS+ channels and is not different from channels neither responding to the stimulus nor showing repetition suppression (SR-RS-). In the β range, SR-RS+ channels also show different β activity compared to SR+RS- channels. Additionally, we found very few sites with negative stimulus response to the first stimulus in a series, suggesting that inhibition is a rare response to the stimuli.”

3. The paradigm employed by the authors confounds adaptation/repetition suppression and expectation suppression which are empirically distinct phenomena, although both might in principle be explainable by predictive coding. The problem is that the repeated stimuli are both predictable and repeated: how can one distinguish an effect of repetition (i.e. repetition suppression/adaptation) from an effect of prediction/expectation? Given this issue, the discussion of their results in terms of predictive coding is highly speculative.

Reply: We agree with the reviewer that our paradigm does not allow to conclusively dissociate between repetition and expectation. However, recent evidence suggests that subjects are less likely to apply overt expectations (even when available) when the stimuli are not task-related, as in our paradigm. We added this to the limitation section of the discussion which now reads:

“In our study we focused on the repeated and hence predictable part of the auditory stimulation. How repetition and expectation suppression interact is still debated. Our paradigm does not allow us to conclusively dissociate between repetition independent of expectation. However, recent evidence [76] suggests that subjects are less likely to apply overt expectations (even when available) when the stimuli are not task-related (as was the case in our study). But recent evidence in the visual modality suggests that when attention is directed elsewhere, expectation vanished despite robust repetition suppression being still evident [76, 77].”

4. The analyses of Figure 2 are interesting but one cannot know from these analyses whether the differences are because of response versus no (or weak) response to the stimulus or repetition suppression versus no repetition suppression since these two factors are confounded/correlated in their analysis: the authors compare responsive channels without repetition suppression with unresponsive channels with repetition suppression. I do not see the purpose of these analyses in the context of the present paper.

Reply: the reviewer is correct that in the previous version our analysis scheme did not allow for spectral differences to be ascribed to the stimulus response or repetition suppression electrodes since we only compared power spectral density in SR+RS- with SR-RS+ channels. In a new analysis we compared PSD values in a two-way ANOVA with the factor channel set (SR+RS-, SR-RS+, SR-RS-, and SR+RS+) and frequency band (θ , α , and β) and found a significant interaction between the factors. In contrast to α activity, we found a significant main effect for channel-type both in the θ and β activity with higher θ power in SR+ channels (SR+RS- and SR+RS+) but higher β power in RS+ channels (SR-RS+ and SR+RS+).

The respective paragraph in the Methods section now reads:

“We estimated the power spectral density (PSD) in each trial (collapsing across all S_1 , S_2 , and S_3 trials) separately for SR+ and RS+ channels using Welch's method based on the FFT [25]. Specifically, for each channel we calculated PSD as a function of frequency (1-30 Hz, 1 Hz steps) in each trial in temporal intervals of 100 ms (due to the short SOA of 600 ms) between -.1 s to .3 s in steps of 50 ms. The resulting

PSD values were averaged across trials yielding one PSD for each channel. We then averaged across three canonical low frequency bands θ (4-8Hz), α (8-12Hz), and β (15-30 Hz) and compared the resulting power estimates across channels using a 2-way ANOVA with the factors channel type (SR+RS-, SR-RS+, SR-RS-, and SR+RS+; see **Fig 3a** for the spectrum of all channel sets separately) and frequency band (θ , α , β). Post-hoc we compared the power estimates between channel sets, separately for the three different frequency bands. To correct for multiple comparisons, p-values were assigned to each t-value within a surrogate distribution constructed by randomly assigning labels (SR+RS-, SR-RS+, SR-RS-, and SR+RS+ individual channels) and corrected by applying the FDR. t-values with a corresponding $q < .05$ (corrected p value) were classified as statistically significant.”

We added the following paragraph to the Results section:

“We then asked whether SR and RS sites dissociate in spectral characteristics. Power-spectral-density (PSD) showed an interaction between factors Channel Type (SR+RS-, SR-RS+, SR-RS-, SR+RS+) and Frequency Bands (θ , α , β) ($F_{6,1224} = 3.2$; $p = .013$; **Fig 3a**). θ activity differed significantly between channels types ($P_{\theta_SR+RS-} = 35.9$, $P_{\theta_SR-RS+} = 33.4$, $P_{\theta_SR-RS-} = 34.3$, $P_{\theta_SR+RS+} = 36.1$; $F_{3,408} = 3.37$; $p = .012$) as well as β activity ($P_{\beta_SR+RS-} = 19.5$, $P_{\beta_SR-RS+} = 21.9$, $P_{\beta_SR-RS-} = 19.1$, $P_{\beta_SR+RS+} = 21.5$; $F_{3,408} = 2.76$; $p = .04$) but not α activity ($P_{\alpha_SR+RS-} = 30.7$, $P_{\alpha_SR-RS+} = 30.2$, $P_{\alpha_SR-RS-} = 30.5$, $P_{\alpha_SR+RS+} = 31.5$; $F_{3,408} = .3$; $p = .82$). Post-hoc test revealed stronger θ power in SR+ than SR- channels ($P_{\theta_SR-RS+} < P_{\theta_SR+RS-}$; $t_{crit} = 1.99$; $t_{106} = 2.51$; $q = .04$; $P_{\theta_SR+RS-} > P_{\theta_SR-RS-}$; $t_{379} = 2.11$; $q = .052$ and a trend towards $P_{\theta_SR-RS+} > P_{\theta_SR+RS+}$; $t_{29} = 1.50$; $q = .082$). In contrast, β activity showed stronger power for RS+ than RS- channels ($P_{\beta_SR-RS+} > P_{\beta_SR+RS-}$; $t_{106} = 2.37$, $q = .0297$; $P_{\beta_SR-RS+} > P_{\beta_SR-RS-}$; $t_{319} = 2.41$, $q = .0297$; and a trend towards $P_{\beta_SR+RS-} < P_{\beta_SR+RS+}$, $t_{89} = 1.5$, $q = .082$ and $P_{\beta_SR-RS-} < P_{\beta_SR+RS+}$, $t_{302} = 1.4$, $q = .084$ **Fig 3b**).”

The corresponding figure now is Fig 3b:

within Fig3:

The caption of Fig3 now reads

“Figure 3: stimulus response and repetition suppression show complex spectral interactions. **a** shows power spectral density (PSD) for SR-RS+ and SR+RS- channels (inset shows PSD for channels showing both/neither effect), the shaded area denotes the standard error across channels. **b** shows spectral power averaged within three canonical frequency bands. We found higher β power in RS+ (SR+RS+, SR-RS+) than RS- (SR+RS-, SR-RS-) channels and different pattern for θ activity with higher θ power in SR+ (SR+RS-, SR+RS+) compared to SR- (SR-RS+, SR-RS-) channels. No differences between channel sets were found within the α band (greyed color bars). **c, d** show HFA modulation by θ (**c**) and β (**d**) for SR-RS+ (blue) and SR+RS- (green) channels, shaded areas denote the standard errors across channels. **e** shows the Kullback-Leibler divergence of θ and β for SR-RS+ and SR+RS-. **f, g** show the modulation of concentration index κ indicating transient θ : β interaction

averaged across phases, the dashed red line shows the significance threshold (*f*) and for each θ phase (*g*) (the shaded area denotes the standard error across channels). *h* shows θ -phase: β -phase coupling at κ_{\max} , gray lines represent pairwise combinations, red dashed line shows average across all pairs *i* shows the κ -comparison between θ : β , θ : α and α : β (bottom: ANOVA F-values), shaded areas denote the standard errors across channels *j* shows Post-hoc comparisons of θ : β , θ : α , and α : β . (***) indicates $p < .001$ *k* shows the directed information propagation from SR+RS- to SR-RS+.”

5. It is close to impossible to fully understand the analyses or Figures without reading the Methods which is presented as the Supplem. Text. This should be remedied. After reading the Suppl. Text on Methods, details of some Figure panels were still unclear.

(a) Are the bands/shading standard errors of the mean or confidence intervals?

Reply: We apologize for this inconvenience. With the extended version, we also expanded the caption with which we believe improves readability. Furthermore, we split the figures in 3 new figures. The new caption of Figure 1 now reads:

“Figure 1: stimulus response and repetition suppression show different spatial profiles. *a* shows the auditory oddball paradigm. While the occurrence of deviants could be unpredictable, S_1 , S_2 and S_3 were always predictable. *b* shows the spatial distribution of HFA SR+RS- (green), and SR+RS+ (magenta) channels. Remaining electrodes are marked by white dots. *c* shows the HFA amplitude modulation of SR+ channels over time averaged across electrodes. The shaded area denotes the standard error across channels. *d* shows the spatial distribution of HFA SR-RS+ channels (blue). Channels with magenta circle represent SR+RS+, as in *b*. *e* Modulation of repetition-suppression (F-values) in SR-RS+ channels. The dashed blue line represents significance threshold of F-values. The shaded area denotes the standard error across channels.”

The new caption of Figure 2 now reads

“Figure 2: stimulus response and repetition suppression show different temporal profiles. *a,b* show the HFA amplitude modulation of SR+RS- (*a*) and SR-RS+ (*b*) channels to S_1 , S_2 and S_3 , the dashed black curve in (*b*) represents the summed sign of differences between S_1 & S_2 and S_2 & S_3 (DiffSR). Positive values of this summed differences indicate a $S_1 > S_2 > S_3$ relationship, shaded areas represent the standard error of the mean. *c, d* show time-resolved BF for modulation of HFA over baseline (evidence for stimulus response) averaged separately across the four different channel sets (SR-RS+, SR+RS-, SR-RS-, and SR+RS+) (*c*) and for the effect of repetition suppression averaged across channels, shaded areas represent the standard error of the mean (*d*). Only SR+RS- and SR+RS+ channels show evidence for a stimulus response, only SR-RS+ and SR+RS+ channels show evidence for RS. *e* shows peak latencies for RS (F-values, blue) and SR (HFA, green), compared within SR+RS+ channels (upper bars) and between SR+RS- and SR-RS+ channels (lower bars), error bars represent the standard error of the mean.”

The new caption of Figure 3 now reads

“Figure 3: stimulus response and repetition suppression show complex spectral interactions. *a* shows power spectral density (PSD) for SR-RS+ and SR+RS- channels (inset shows PSD for channels showing both/neither effect), the shaded area denotes the standard error across channels. *b* shows spectral power averaged within three canonical frequency bands. We found higher β power in RS+ (SR+RS+, SR-RS+) than RS- (SR+RS-, SR-RS-) channels and different pattern for θ activity with higher θ power in SR+ (SR+RS-, SR+RS+) compared to SR- (SR-RS+, SR-RS-) channels. No differences between channel sets were found within the α band (greyed color bars). *c,d* show HFA modulation by θ (*c*) and β (*d*) for SR-RS+ (blue) and SR+RS- (green) channels, shaded areas denote the standard errors across channels. *e* shows the Kullback-Leibler divergence of θ and β for SR-RS+ and SR+RS-. *f,g* show the modulation of concentration index κ indicating transient θ : β interaction averaged across phases, the dashed red line shows the significance threshold (*f*) and for each θ phase (*g*) (the shaded area denotes the standard error across channels). *h* shows θ -phase: β -phase coupling at κ_{\max} , gray lines represent pairwise combinations, red dashed line shows average across all pairs *i* shows the κ -comparison between θ : β , θ : α and α : β (bottom: ANOVA F-values), shaded areas denote the standard

errors across channels j shows Post-hoc comparisons of $\theta:\beta$, $\theta:\alpha$, and $\alpha:\beta$. (***) indicates $p < .001$) k shows the directed information propagation from SR+RS- to SR-RS+.”

(b) The F values that are shown in Figure 1 and Figure 2 cannot be F values from an ANOVA since such F values cannot be negative. Also, if F values are close to 0 then typically the model assumptions are not met. Are the shown “F values” baseline corrected (which does not make sense to me)? What does the stippled horizontal line correspond to? The latter also holds for other panels of Figures 1 and 2 where horizontal lines are presented but left unexplained.

Reply: We apologize for this oversight. When we put all figures together, we changed the font and unfortunately displaced the y-axis. We double-checked this issue and now show the correct y-axis. In the baseline interval the F-values vary around 1. The stippled line gives the confidence interval indicating statistical significance. Thanks for catching this.

(c) What is diffSR in Fig 1h?

Reply: We tested in RS+ channels for a monotonical increase. As we described in the methods section, we computed the amplitude decrease over a train of tones across these channels. We averaged mean responses following S1, S2, and S3 across RS+ channels. We then calculated the differences between those responses (response following S3 vs response following S2 and response following S2 vs response following S1, respectively) at each time point and summed the sign function of differences (-1 and +1 for negative and positive differences, respectively).

$$DiffSR = sgn(S_1 - S_2) + sgn(S_2 - S_3)$$

Positive values of $DiffSR$ indicate a $S_1 > S_2 > S_3$ relationship. Higher positive values of $DiffSR$ indicate stronger differences between S1, S2, and S3.

.

We added the missing information to the caption of the respective figure. We hope that the new longer version improves this (Fig 1h is now Fig 2b). The caption now reads

“the dashed black curve in (b) represents the summed sign of differences between S1 & S2 and S2 & S3 ($Diff_{SR}$). Positive values of this summed differences indicate a $S_1 > S_2 > S_3$ relationship, shaded areas represent the standard error of the mean.”

6. Also in the text, clarifications are needed: line 76: what does the Greek symbol indicate?

Reply: In this paragraph we compare baseline variances to rule out the possibility that differences in SR are due to differences in baseline variance. We denoted the variance by ζ^2 but did not make it clear enough in the text. We changed this. The paragraph now reads:

“115 channels exhibited SR and/or RS. Of these only seven channels showed both (designated SR+RS+). The remaining showed only SR (SR+RS-, 84 channels) or showed RS without SR (SR-RS+; 24 channels). We first confirmed that the lack of SR in SR-RS+ sites is not due to reduced sensitivity caused by high baseline variance (ζ^2) in these channels compared to other sites (baseline -200 – 0ms, $F_{(3,408)} = .37$; $p = .78$; SR+RS- : mean $\zeta^2 = .0012$, std = .004; SR-RS+ : $\zeta^2 = .0008$, std = .0008; SR+RS+ : $\zeta^2 = .0017$, std = .002; $\zeta^2 = .0009$, std = .0037).”

Line 83: what is t_{crit} ?

Reply: t_{crit} denotes the critical t value which the observed t values has to exceed to be considered significant.

We added the missing explanation in the paragraph on data analysis in IV – Comparison of dominant band power . It reads now:

“Post-hoc we compared the power estimates between channel sets, separately for the three different frequency bands by computing t-values and comparing those to the critical t_{crit} . t_{crit} denotes the critical t value which the observed t values has to exceed to be considered significant. To correct for multiple comparisons, p-values were assigned to each t-value within a surrogate distribution constructed by

randomly assigning labels (SR+RS⁻, SR⁻RS⁺, SR⁻RS⁻, and SR+RS⁺ individual channels) and corrected by applying the FDR. The adjusted p values are labeled q. T-values with a corresponding $q < .05$ (corrected p value) were classified as statistically significant.”

7. Methods: (a) Why did the authors not use a sign test or Wilcoxon signed-rank test to test for significant responses (paired, trialwise comparison of baseline and stimulus period)? The method they employ now by shifting in a circular way trial times seems to be needlessly complicated.

Reply: We sought to distinguish between random background modulations in the signal from significant stimulus responses. Our approach allows to define the exact onset of time series effects and similar to the Wilcoxon test does not require normality and sphericity in the data. The logic behind this is that the surrogate distribution is generated under the original sample’s distributional and sphericity properties. Furthermore, we don’t agree that the shifting method is complicated. In essence it estimates the parameters as in the original data set with the exception that the original data are shifted in time. This is done with `circshift.m` function in Matlab in a single step.

(b) how was the FDR applied? Taking into account the number of channels and number of time points? Please clarify?

Reply: We apologize for the missing information. We applied the FDR across all channels and time points. In the new long version, we expand the methods section in which we now explain the specific FDR procedure.

The respective part in the methods section now reads:

“The comparison of observed difference values with the surrogate distribution results in a p-value for each channel and time interval. To control for multiple comparisons, we corrected the p-values by applying the false discovery rate (FDR) [21] method across all channels and time intervals. Channels with a $q < .05$ (where q is the false discovery rate) in any of the five intervals were classified as showing a significant HFA modulation following the standard stimuli and were denoted as SR⁺, whereas the remaining channels were labeled as SR⁻.”

(c) Did the authors observe repetition enhancement, i.e. positive r between trial number and response?

Reply: We limited our analyses only to RS effects since HFA RE was observed in only 3 channels which are even less than SR+RS⁺ channels. This ratio of RS and RE sites is in line with previous studies (see for example Korzeniewska et al. [1])

Korzeniewska A, Wang Y, Benz HL et al. (2020) Changes in human brain dynamics during behavioral priming and repetition suppression. *Prog Neurobiol* 189:101788.
<https://doi.org/10.1016/j.pneurobio.2020.101788>

We therefore added the following sentence to the Methods section:

“In contrast, channels with temporal intervals of significant F-value and a significant positive r -value show repetition enhancement (RE+)”

And the results section

“We found 3 channels showing RE (both a significant $F_{N\text{ standard}}$ and significant positive r -value), in the parietal (N=2) and the temporal (N=1) cortex. None of these channels showed significant SR (i.e., they were SR-RE⁺ channels).”

(d) How did they compute the Bayes factor?

Reply: we estimated BF for a modulation over a joint baseline using the bf-toolbox in Matlab (<https://klabhub.github.io/bayesFactor/>) separately for each channel. The baseline comprises all trials and time points within the baseline interval.

We added this missing information to the methods section. The respective paragraph now reads:

“To determine the amount of evidence for a change over baseline we compared HFA values at each time point with HFA values in the baseline interval separately for each channel across trials, using Bayes factor (BF; bf.m toolbox in MATLAB <https://klabhub.github.io/bayesFactor/>). $BF > 3$ is considered strong evidence for a difference (difference is 3 times more likely than no difference) and $BF < 1/3$ supports null effects [22, 23].”

Reviewer #2 (Remarks to the Author):

Ten pre-surgery patients monitored with subdural electrodes heard a long sequence of sounds while watching a slideshow. In blocks of 400 sounds 80% were sound A and 20% were sound B, and within each block sequences were either pseudorandom or a regular train of four As followed by a B. Recordings of high frequency activity were characterised as stimulus response (SR, indicating stimulus evoked amplitude modulation of the response) or repetition suppression (RS, indicating a reduction in the amplitude between successive repeats of a tone) on each recording channel. Of 412 recording sites, 115 responded to the auditory stimulus. Of these 84 were classed as SR but not RS and 24 were classed as RS but not SR. The remaining 7 were classed as both SR and RS. The analysis supports a general dissociation of these two response characteristics between recording sites and between frequency bands. It also suggests differences in the timing of the SR and RS components of the response in sites classed as both SR (for which stimulus modulation was evident in the earlier component) and RS (for which attenuation was evident in the later component). The authors consider these dissociations as evidence that challenges the predictive coding framework.

Overall, I found the study to report a thorough analysis of an interesting and important dataset carried out to a high standard. I consider the conclusions to be supported by the results/analysis. And the implications of the data in terms of the predictive coding framework make this paper a welcome contribution that will help advance the understanding of this important topic, and that of repetition suppression more generally.

However the manuscript is unnecessarily difficult to read in its present format. It is very short, and begins with a results section without identifying/explaining the research question. I had to read the supplementary materials carefully before I could really understand the main text, not least due to the dense packing of abbreviated terms, several of which were undefined (see minor comments). I therefore recommend the following revisions.

1. Include an introduction section that identifies the specific research question in terms of testing the predictions/assumptions of the predictive coding theory, and which motivates the specific design of the study as an appropriate way of carrying out that investigation. This section could also emphasise the novelty of this dataset (human electrode recordings) in the context of other primate single-unit recording studies of repetition suppression (e.g., Li L, Miller EK, Desimone R. The representation of stimulus familiarity in anterior inferior temporal cortex. *Journal of Neurophysiology*. 1993; 69(6):1918–1929.)

Reply: We agree with the reviewer regarding the difficulty of the short format, given the complex analyses. Hence, we created a longer, more extensive version and we included an introduction that leads to the specific issue based on the theoretical framework of Repetition Suppression and the model of Predictive Coding, also emphasizing the critical features (including human electrode recording as mentioned by the reviewer, also in the context of existing single-unit studies like the one mentioned).

The first paragraph of the new introduction now reads:

“A ubiquitous finding in neuroscience is that neural responses to repeated stimuli are reduced compared to initial stimulus presentation, the phenomenon of repetition suppression (RS). RS has been shown in both single-unit studies in the monkey cortex [1–3] as well as noninvasive studies in humans using different techniques (for a review, see [4]). Several explanations for RS have been put forward, including adaptation or habituation, sharpening of representations, and reduction of prediction errors [4–12]. This reduction of responses to frequently occurring stimuli is associated with an enhanced response to unexpected events, establishing a mechanism for change detection [13, 14], with the probability of stimulus events accounting for a large proportion of neural variability [9]. Most hypotheses on the mechanisms responsible for RS, assume that what is suppressed is the stimulus-induced response. That is, the same neurons or networks that are initially responsive to the stimulus are the ones which are less active when the same stimulus repeats. Non-invasive studies in humans report that RS and stimulus-response (SR) overlap but these methods cannot distinguish nearby cortical activity.

A critical question remains whether RS is restricted to reducing the SR, in which case SR and RS should co-occur in the same electrodes (henceforward SR+RS+ sites), as suggested by scalp EEG and MEG recordings. Alternatively, SR and RS could be dissociated but the circuits exhibiting SR and RS are intermingled and not resolvable by low resolution scalp recording. This potential dissociation could be measured using direct cortical recording of broad band high frequency activity (HFA, 80-150Hz), which is the key response frequency in previous ECoG (electrocorticography) studies [15–19] studying SR and RS.

Here, we utilized the high temporal and spatial resolution of direct cortical recordings from subdural ECoG electrodes to compare SR+ and RS+ signals in ten patients, presented with trains of task-irrelevant auditory stimuli, while attending a visual slide show, to probe the automatic nature of RS. We show that while SR and RS both engage frontal, parietal and temporal regions, they can be dissociated temporally and spatially in HFA. Critically, HFA SR+ and RS+ sites are distinctly modulated by θ and β low frequency activity, respectively, with mutual information flow from SR+ to RS+ sites.”

2. Incorporate the bulk of the supplement (particularly the analysis subsections) into a methods section in the main text so that the manuscript explains the various quantities that are subjected to analysis prior to presenting the results of those analyses.

Reply: Following the reviewers’ suggestion, in the current longer version of the manuscript, we also integrated our methods part into the manuscript directly – with identical subheadings as results section to enable quickly reading specific data analysis steps and their according results.

3. Improve the clarity of the figures.

3a. Figure 1: The authors should add a descriptive title that summarises the take-home point of the figure. The text in the figure is unreadable in places and should be enlarged throughout. Presumably the top and bottom traces in a depict the regular and pseudorandom blocks (please clarify with a label)? Please state what Diff_{SR} means in the caption for panel h and also label the righthand y-axis. Consider adding labels SR+RS– and SR–RS+ to panels i and j. I would recommend unpacking some of the abbreviations in the caption to make it more readable.

3b. Figure 2: The authors should add a descriptive title that summarises the take-home point of the figure, and increase the size of the text throughout. Some of the labels, e.g., ‘phase:phase frequency’ are difficult to understand given that they are explained in the supplement rather than the main text.

Reply: Following the suggestion of the reviewer, we changed the structure of our figures, separating them in three figures. Therefore, we grouped the figures according to difference between SR+RS– and SR–RS+ channels as to their spatial (Figure 1) and temporal (Figure 2) profiles and their spectral interaction (Figure 3). Each figure starts now with a concise take-home message. In all figures, we added a title, increased font size and added the missing labels. As we now included an extensive methods part in the text, rather than in a separate summary, we hope to have improved readability of specific labels within the figures.

The summary is for Fig 1: “***Figure 1: stimulus response and repetition suppression show different spatial profiles***”

In Fig1a (showing the tones used in the paradigm) we added labels of regular and pseudorandom trials. Fig 1h is now Fig 2b, in which we improved the caption, that now reads: “***a,b*** show the HFA amplitude modulation of SR+RS– (***a***) and SR–RS+ (***b***) channels to S_1 , S_2 and S_3 , the dashed black curve in (***b***) represents the summed sign of differences between S_1 & S_2 and S_2 & S_3 (Diff_{SR}). Positive values of this summed differences indicate a $S_1 > S_2 > S_3$ relationship, shaded areas represent the standard error of the mean.”

We added a right-hand y-axis label. On top, we merged Fig 1i and j to one figure, which is now Fig 2e. Here, we also added labels within the figure to state each channel population (SR+RS- etc).

With the longer version, we also expanded the caption with which we hope to improve the readability. Furthermore, we split the figures in 3 new figures. The new caption of Figure 2 now reads:

“Figure 2: stimulus response and repetition suppression show different temporal profiles. a, b show the HFA amplitude modulation of SR+RS- (**a**) and SR-RS+ (**b**) channels to S₁, S₂ and S₃, the dashed black curve in (**b**) represents the summed sign of differences between S₁ & S₂ and S₂ & S₃ (Diff_{SR}). Positive values of this summed differences indicate a S₁>S₂>S₃ relationship, shaded areas represent the standard error of the mean. **c, d** show time-resolved BF for modulation of HFA over baseline (evidence for stimulus response) averaged separately across the four different channel sets (SR-RS+, SR+RS-, SR-RS-, and SR+RS+) (**c**) and for the effect of repetition suppression averaged across channels, shaded areas represent the standard error of the mean (**d**). Only SR+RS- and SR+RS+ channels show evidence for a stimulus response, only SR-RS+ and SR+RS+ channels show evidence for RS. **e** shows peak latencies for RS (F-values, blue) and SR (HFA, green), compared within SR+RS+ channels (upper bars) and between SR+RS- and SR-RS+ channels (lower bars), error bars represent the standard error of the mean.”

The new caption of Figure 3 now reads

“Figure 3: stimulus response and repetition suppression show complex spectral interactions. a shows power spectral density (PSD) for SR-RS+ and SR+RS- channels (inset shows PSD for channels showing both/neither effect), the shaded area denotes the standard error across channels. **b** shows spectral power averaged within three canonical frequency bands. We found higher β power in RS+ (SR+RS+, SR-RS+) than RS- (SR+RS-, SR-RS-) channels and different pattern for θ activity with higher θ power in SR+ (SR+RS-, SR+RS+) compared to SR- (SR-RS+, SR-RS-) channels. No differences between channel sets were found within the α band (greyed color bars). **c, d** show HFA modulation by θ (**c**) and β (**d**) for SR-RS+ (blue) and SR+RS- (green) channels, shaded areas denote the standard errors across channels. **e** shows the Kullback-Leibler divergence of θ and β for SR-RS+ and SR+RS-. **f, g** show the modulation of concentration index κ indicating transient θ : β interaction averaged across phases, the dashed red line shows the significance threshold (**f**) and for each θ phase (**g**) (the shaded area denotes the standard error across channels). **h** shows θ -phase: β -phase coupling at κ_{\max} , gray lines represent pairwise combinations, red dashed line shows average across all pairs **i** shows the κ -comparison between θ : β , θ : α and α : β (bottom: ANOVA F-values), shaded areas denote the standard errors across channels **j** shows Post-hoc comparisons of θ : β , θ : α , and α : β . (***) indicates $p < .001$) **k** shows the directed information propagation from SR+RS- to SR-RS+.”

Minor comments:

4. Line 50 - SR+RS+ — it would help clarify that this is the authors’ label for the condition where RS and SR overlap (e.g., use ‘designated’ as in later examples).

Reply: We agree with the reviewer, this sentence now reads “Of these only seven channels showed both (designated SR+RS+)”.

5. Line 54 - S_1-S_3 — on first reading this appears to be a subtraction of two quantities, rather than a reference to three repeats of a given stimulus sound, please clarify.

Reply: We agree with the reviewer that this is misleading. We changed all instances where this occurred.

For example, the caption of Figure 1 now reads:

Figure 1: stimulus response and repetition suppression show different spatial profiles. a shows the auditory oddball paradigm. While the occurrence of deviants was unpredictable, S₁, S₂ and S₃ were always predictable. **b** shows the spatial distribution of HFA SR+RS- (green), and SR+RS+ (magenta)

channels. Remaining electrodes are marked by small white dots. *c* shows the HFA amplitude modulation of SR+ channels over time averaged across electrodes. The shaded area denotes the standard error across channels. *d* shows the spatial distribution of HFA SR–RS+ channels (blue). Channels with magenta circle represent SR+RS+, as in *b*. *e* Modulation of repetition-suppression (F-values) in SR–RS+ channels. The dashed blue line represents significance threshold of F-values. The shaded area denotes the standard error across channels.

6. Line 56 - S1 should be S₁ and S3 should be S₃? Also RS should be described as a significant reduction of activity from S₁ to S₂ and from S₂ to S₃, to reflect the calculations described in the supplement.

Reply: We agree that the writing was not consistent. We now write S₁, S₂, S₃ instead of S1, S2, S3. In the new version we described in the methods section that we tested for a monotonical decrease which is also shown in Fig 1.

The respective paragraph reads:

“In addition, to ensure that RS+ show a S₁>S₂>S₃ relationship, we computed the amplitude decrease over a train of tones across these channels. We averaged mean responses following S₁, S₂, and S₃ across RS+ channels. We then calculated the differences between those responses (response following S₃ vs response following S₂ and response following S₂ vs response following S₁, respectively) at each time point and summed the sign function of differences (-1 and +1 for negative and positive differences, respectively).

$$diffSR = \text{sgn}(S_1 - S_2) + \text{sgn}(S_2 - S_3)$$

Positive values of *DiffSR* indicate a S₁>S₂>S₃ relationship. Higher positive values of *DiffSR* indicate stronger differences between S₁, S₂, and S₃. “

7. Line 52 and Supplement line 31 - please define ECoG. I assume you mean electrocorticography but I doubt the general reader will be familiar.

Reply: We now defined ECoG as electrocorticography both in the abstract:

“We challenge this conjecture using electrocorticographic (ECoG) recordings with high spatial resolution”

and in the main text (intro):

“This potential dissociation could be measured using direct cortical recording of broad band high frequency activity (HFA, 80-150Hz), which is the key response frequency in previous ECoG (electrocorticography) studies [15–19] studying SR and RS.”

8. Supplement line 49 - please define HFA. It is defined only in the abstract.

Reply: We now defined HFA as high frequency activity both in the abstract:

“SR and RS were indexed by high-frequency activity (HFA) across temporal, parietal, and frontal cortices”

and in the main text (intro):

“This potential dissociation could be measured using direct cortical recording of broad band high frequency activity (HFA, 80-150Hz), which is the key response frequency in previous ECoG (electrocorticography) studies [15–19] studying SR and RS.”

9. Supplement line 87 - monotonously should be monotonically?

Reply: We changed all instances of monotonous to monotonical and monotonously to monotonically, respectively.

For example the first paragraph of the methods section in which we describe the Repetition Suppression now reads:

“RS is defined as attenuated amplitude to repeated stimulus presentation. Hence, this definition is two-fold: (i) a change of amplitude which is (ii) monotonically decreasing with the number of repeated stimulus presentations. While (i) refers to statistical differences in brain response with repetitions, (ii) assumes a specific model of response attenuation. We thus grouped trials according to the number of standards in a train in three groups (S_1 , S_2 , and S_3) since only the first three standards in a train can be expected in both conditions. To parameterize the amplitude modulation with stimulus repetition (i), we ran a one-way ANOVA with factor Number of Standards for each electrode (with trials as random variable), regardless of whether it was SR+ or SR-, at every time point, both in the regular and irregular condition. This leads to an F-value time series (main effect: $F_{N_{\text{standard}}}$) for each channel in each condition. Significant F-values only define differences between numbers of preceding standards but not the exact model of monotonical decrease of neural responses. We tested (ii) the model of a monotonic neural amplitude decrease across the number of repetitions”

10. Supplement line 105 - please define ‘the FDR procedure’

Reply: With the new longer version we introduce the abbreviation FDR in the methods section which stands for false discovery rate. The respective part now reads:

“The comparison of observed difference values with the surrogate distribution results in a p-value for each channel and time interval. To control for multiple comparisons, we corrected the p-values by applying the false discovery rate (FDR) [21] method across all channels and time intervals.”

11. Supplement line 93 - please give a definition for SR- in the supplement.

Reply: We integrated the supplement into the main text and added a formal definition of SR+ and SR- at the end of the subsection “Stimulus Response” within the methods section, which now reads

“Channels with a $q < .05$ (where q is the false discovery rate) in any of the five intervals were classified as showing a significant HFA modulation following the standard stimuli and hence were denoted as SR+, whereas the remaining channels were labeled as SR-”

12. Supplement line 147 - closing bracket missing.

Reply: We added a closing bracket.

13. Supplement line 156 - should RS be RS+ here?

Reply: The reviewer is right, indeed it should be RS+, we added this to the methods section. The respective sentence now reads:

“The resulting HFA histograms – each containing 50 values – were averaged separately for SR+ and RS+, separately for θ and β activity.”

14. Supplement line 160 - ‘parameterizes’ should instead be e.g., ‘measures’ or ‘describes’ etc.

Reply: We replaced “parameterizes” with “describes”. The sentence now reads: “The interaction effect of the ANOVA describes the double dissociation of HFA of SR+ and RS+ PAC to θ and β networks, respectively.”

15. Finally, the authors may wish to take a look at a recent paper of mine in which it is suggested that repetition suppression may reflect the sharpening of neuronal receptive fields via plasticity mediated by

recurrent inhibitory interactions. Please feel under no obligation at all to cite this paper - I mention it because the reported temporal dissociation of SR and RS at the SR+RS+ sites seems to me to be consistent with a role of lateral inhibition in attenuating the neuronal responses to repeated stimuli: Spigler G, Wilson SP (2017) Familiarization: A theory of repetition suppression predicts interference between overlapping cortical representations. PLoS ONE 12(6): e0179306. doi:10.1371/journal.pone.0179306 Stuart Wilson

Reply: We thank the reviewer for this interesting idea and the mentioned paper. Indeed, the idea of interpreting the dissociation of SR and RS as, as seen by the Sharpening theory, as distinct processes of afferent weighted connections and lateral weighted connections is promising. We added a new paragraph to the discussion which now reads:

“RS can be explained by neural sharpening [4, 12, 43, 44] due to fall-out of neurons that are not optimally tuned to stimulus features with repetition. Repeated sensory evidence strengthens intracortical inhibitory connections. This lateral interaction [45] may cause a decrease of the population response (‘inhibitory sharpening’, [12]). Even though we cannot directly test neural sharpening, lateral interactions can be an explanation for the SR+RS– vs SR–RS+ sites. This sharpening may also be influenced by top-down inhibition as a component of hierarchical predictive coding.”

Reviewer #3 (Remarks to the Author):

In this manuscript, the authors studied stimulus-response and repetition-suppression with ECoG. Their main finding is that stimulus-response and repetition-suppression dynamics revealed temporally and spatially dissociated but interacting neuronal activity. This is a highly novel and exciting finding in the field. The discovery will undoubtedly form the basis for several future studies and enlighten the previous results on repetition-suppression. The conclusions of the manuscript are original. Overall, the manuscript is well-written, logical, and scientifically rigorous. I only have a little comment. The authors should define the abbreviation **PC**.

Reply: We thank the reviewer very much for the positive evaluation of our manuscript. We now define the abbreviation PC both in the abstract (this sentence here now reads “In contrast to predictive coding (PC) accounts our results indicate that HFA_{SR} and HFA_{RS} are functionally linked but have minimal spatial overlap.”) and in the main text (this sentence here now reads “Predictive coding (PC) schemes mostly assume that the stimulus response itself is suppressed when it is predicted, indicating that the evaluation of a stimulus likelihood precede or take place simultaneously with the bottom up response to the stimulus, so that only deviations from prediction are registered”).

REVIEWERS' COMMENTS:

Reviewer #1 (Remarks to the Author):

The authors replied to all my concerns and revised the paper thoroughly, with new analyses. I have one remaining comment regarding their analysis of the power in different frequency bands (new Figure 3b). They averaged across the 3 trial types (S1, S2, and S3). Hence, the difference between the 4 response profiles between the beta and gamma bands could be due to overall response magnitude and/or a difference in the impact of repetition on the power in these frequency bands. They should analyze the power for the three trial types separately to distinguish these two factors, which may change the interpretation of this result.

I have one minor comment regarding their new Figure 2: please explain what the three horizontal lines indicate in Figures 2c and 2d.

Reviewer #2 (Remarks to the Author):

I appreciate the efforts of the authors to substantially re-organise the paper, which is now far more readable and understandable. All of my previous comments have been satisfactorily addressed. On re-reading the paper I found the following typos that the authors may wish to correct.

Stuart Wilson

- The Procedure section of the Methods ends with (...) and should be fixed.
- Data analysis section IV, red text "...values has to exceed..." should be "...values had to exceed...".
- Data analysis section VI, "ranging from $-\pi$ to πS ", the "S" should be removed.
- Data analysis section VI "To correct for multiple comparisons, p-value were assigned" ("p-value" should be "p-values")
- Results opening sentence, I suggest "easier" be changed to "easy".
- Results section I, "Figure" should be changed to "Fig" for consistency
- Results section II, typo "aveage".
- Results section IV, typo "...between channels types..." should be "...between channel types..."
- Results section IV, formatting issue with closing bracket on "(P_\theta)"
- Results section IV, "values has to exceed" should be "values had to exceed"
- Results section IV, semi-colon missing before "Fig 3b".
- Results section VI, "values has to exceed" should be "values had to exceed"
- Discussion, second paragraph of red text, I suggest changing "option" to "possibility"
- Discussion, third paragraph of red text, should end with a full stop.
- Discussion, paragraph starting "The SR+ and RS+ ensembles...", the second sentence does not end properly.
- Discussion, penultimate paragraph, suggest changing character "2" to word "two".
- Figure 1 caption typo "circle" should be "circles" and "the" missing from final sentence.
- Figure 3e - it is unclear what is the difference between the + symbol and the * symbol to indicate statistical differences?
- Figure 3 caption, "...channels and different pattern..." should be "...channels and a different pattern..."

Reviewer #1 (Remarks to the Author):

The authors replied to all my concerns and revised the paper thoroughly, with new analyses. I have one remaining comment regarding their analysis of the power in different frequency bands (new Figure 3b). They averaged across the 3 trial types (S1, S2, and S3). Hence, the difference between the 4 response profiles between the beta and gamma bands could be due to overall response magnitude and/or a difference in the impact of repetition on the power in these frequency bands. They should analyze the power for the three trial types separately to distinguish these two factors, which may change the interpretation of this result.

Reply: we are grateful to the reviewer for raising this point, allowing us to inspect the dynamics of the different bands across repetitions. We repeated the PSD analysis again for SR–RS+ and SR+RS– channels in the theta and beta bands, separately for S1, S2 and S3. First, we found that spectral power remains at a comparable level within both channel sets and frequency bands for S1 through S3, and the pattern we describe for the average across S1-S3 is present for each of the stimuli separately (see new Figure 3b). That is, theta power to S1, S2, and S3 is lower in SR–RS+ channels than theta power to S1, S2, and S3 in SR+RS–.

New figure 3:

The caption now reads:

Figure 3: stimulus response and repetition suppression show complex spectral interactions. **a** power spectral density (PSD) for SR-RS+ and SR+RS- channels (inset shows PSD for channels showing both/neither effect). **b** spectral power averaged within three canonical frequency bands, showing higher β power in RS+ (SR+RS+, SR-RS+) than RS- (SR+RS-, SR-RS-) channels and higher θ power in SR+ (SR+RS-, SR+RS+) compared to SR- (SR-RS+, SR-RS-) channels. No differences between channel sets were found within the α band (unsaturated color bars). The right panel shows spectral power for the 3 consecutive standard trials separately, for the SR-RS+ channels (blue) and SR+RS- channels (green) separately. **c,d** HFA modulation by θ (**c**) and β (**d**) for SR-RS+ (blue) and SR+RS- (green) channels. **e** Kullback-Leibler divergence of θ and β for SR-RS+ and SR+RS-. **f,g** modulation of concentration index κ , indicating transient θ : β interaction, averaged across phases (**f**, the dashed horizontal red line shows the significance threshold) and for each θ phase (**g**). **h** θ -phase: β -phase coupling at κ_{max} , gray lines represent pairwise combinations, red dashed line shows average across all pairs. **i** the κ -comparison between θ : β , θ : α and α : β (bottom: ANOVA F-values). **j** Post-hoc comparisons of θ : β , θ : α , and α : β . (***) indicates $p < .001$) **k** directed information propagation from SR+RS- to SR-RS+. In all panels, shaded error margin around lines indicate the standard error across channels.

Second, for SR–RS+ channels we found a trend for change in spectral power with repetition. Theta power increased numerically from S1 through S3 while beta power decreased. This can be seen in figure 3b and across all frequencies in the figure below where the blue and orange line show the summed differences at each frequency between S1 vs. S2 and S2 vs. S3 for SR+RS– and SR–RS+ channels, respectively. Positive values denote an increase (S1<S2<S3, i.e. repetition enhancement) while negative values correspond to a decrease (S1>S2>S3, i.e. repetition suppression). SR+RS– channel showed neither RE nor RS (almost 0 difference in power in SR+RS– channels across the entire spectrum in the figure below). Our main results in this paper show that SR–RS+ channels had higher β power than SR+RS– channels, and this turns out to be in spite of the fact that these channels show a small reduction in β power with repetition (repetition suppression). The reverse pattern we observed in the θ range, where the SR+RS– channels show stronger power than SR–RS+ channels even though the latter exhibit repetition enhancement.

The respective results section now reads:

“We then asked whether SR and RS sites dissociate in spectral characteristics. Power-spectral-density (PSD) showed an interaction between factors Channel Type (SR+RS–, SR–RS+, SR–RS–, SR+RS+) and Frequency Bands (θ , α , β) ($F_{6,1224} = 3.2$; $p = .013$; **Fig 3a**). θ and β activity differed significantly between channel types but not α activity, see **Table 2**. Post-hoc tests revealed stronger θ power (P_θ) in SR+ than SR– channels ($P_{\theta_SR-RS+} < P_{\theta_SR+RS-}$; $t_{106} = 2.51$; $q = .04$; $P_{\theta_SR+RS-} > P_{\theta_SR-RS-}$; $t_{379} = 2.11$; $q = .052$ and a trend towards $P_{\theta_SR-RS+} > P_{\theta_SR+RS+}$; $t_{29} = 1.50$; $q = .082$; $t_{crit} = 1.99$ denotes the t value which the observed t values had to exceed to be considered significant). In contrast, β activity showed stronger

power for RS+ than RS- channels ($P_{\theta_{SR-RS+}} > P_{\beta_{SR+RS-}}$, $t_{106} = 2.37$, $q = .0297$; $P_{\theta_{SR-RS+}} > P_{\theta_{SR-RS-}}$, $t_{319} = 2.41$, $q = .0297$; and a trend towards $P_{\beta_{SR+RS-}} < P_{\beta_{SR+RS+}}$, $t_{89} = 1.5$, $q = .082$ and $P_{\beta_{SR-RS-}} < P_{\beta_{SR+RS+}}$, $t_{302} = 1.4$, $q = .084$; **Fig 3b**). These results show higher β power in SR-RS+ channels than SR+RS- channels, – despite a small reduction in β power from S₁ through S₃, and stronger θ power in SR+RS- channels show than SR-RS+ channels even though the latter exhibit numerically repetition enhancement. (see **Fig 3b** right panel).

I have one minor comment regarding their new Figure 2: please explain what the three horizontal lines indicate in Figures 2c and 2d.

Reply.

We apologize for the missing information. The gray lines denote the critical Bayes levels 3, 1, and .3. Values exceeding 3 indicate evidence for the alternative hypothesis and values below .3 indicate evidence for the null hypothesis.

The caption now reads:

Figure 2: stimulus response and repetition suppression show different temporal profiles. *a,b* show the HFA amplitude modulation of SR+RS- (*a*) and SR-RS+ (*b*) channels to S₁, S₂ and S₃, the dashed black curve in (*b*) represents the summed sign of differences between S₁ & S₂ and S₂ & S₃ (Diff_{SR}). Positive values of this summed differences indicate a S₁>S₂>S₃ relationship, shaded areas represent the standard error of the mean. *c, d* show time-resolved BF for modulation of HFA over baseline (evidence for stimulus response) averaged separately across the four different channel sets (SR-RS+, SR+RS-, SR-RS-, and SR+RS+) (*c*) and for the effect of repetition suppression averaged across channels, shaded areas represent the standard error of the mean (*d*). Only SR+RS- and SR+RS+ channels show evidence for a stimulus response, only SR-RS+ and SR+RS+ channels show evidence for RS. The gray horizontal lines correspond to Bayes values 3, 1, and .3 with 3 and .3 denoting the critical levels for evidence for alternative and null hypothesis, respectively. *e* shows peak latencies for RS (F-values, blue) and SR (HFA, green), compared within SR+RS+ channels (upper bars) and between SR+RS- and SR-RS+ channels (lower bars), error bars represent the standard error of the mean.

Reviewer #2 (Remarks to the Author):

I appreciate the efforts of the authors to substantially re-organise the paper, which is now far more readable and understandable. All of my previous comments have been satisfactorily addressed. On re-reading the paper I found the following typos that the authors may wish to correct.

Stuart Wilson

- The Procedure section of the Methods ends with (...) and should be fixed.
- Data analysis section IV, red text "...values has to exceed..." should be "...values had to exceed...".
- Data analysis section VI, "ranging from -pi to piS", the "S" should be removed.
- Data analysis section VI "To correct for multiple comparisons, p-value were assigned" ("p-value" should be "p-values")
- Results opening sentence, I suggest "easier" be changed to "easy".
- Results section I, "Figure" should be changed to "Fig" for consistency
- Results section II, typo "aveage".
- Results section IV, typo "...between channels types..." should be "...between channel types..."
- Results section IV, formatting issue with closing bracket on "(P_\theta)"
- Results section IV, "values has to exceed" should be "values had to exceed"
- Results section IV, semi-colon missing before "Fig 3b".
- Results section VI, "values has to exceed" should be "values had to exceed"
- Discussion, second paragraph of red text, I suggest changing "option" to "possibility"
- Discussion, third paragraph of red text, should end with a full stop.
- Discussion, paragraph starting "The SR+ and RS+ ensembles...", the second sentence does not end properly.
- Discussion, penultimate paragraph, suggest changing character "2" to word "two".
- Figure 1 caption typo "circle" should be "circles" and "the" missing from final sentence.
- Figure 3e - it is unclear what is the difference between the + symbol and the * symbol to indicate statistical differences?
- Figure 3 caption, "...channels and different pattern..." should be "...channels and a different pattern..."

We thank the reviewer very much for the thorough evaluation of the manuscript. We edited all of the above-mentioned issues in the new version of the manuscript.